# Lipidomic biomarkers in plasma correlate with disease severity in adrenoleukodystrophy
Yorrick R. J. Jaspers [1], Hemmo A. F. Yska[2], Caroline G. Bergner[3], Inge M. E. Dijkstra[1], Irene C. Huffnagel[2], Marije M. C. Voermans[2], Eric Wever [4,5], Gajja S. Salomons[1,5,6], Frédéric M. Vaz [1,5], Aldo Jongejan [4], Jill Hermans[1,5], Rebecca K. Tryon [7], Troy C. Lund [7], Wolfgang Köhler [3], Marc Engelen[2,8] & Stephan Kemp [1,8] ✉

## Abstract

**Background** X-linked adrenoleukodystrophy (ALD) is a neurometabolic disorder caused by pathogenic variants in ABCD1 resulting very long-chain fatty acids (VLCFA) accumulation in plasma and tissues. Males can present with various clinical manifestations, including adrenal insufficiency, spinal cord disease, and leukodystrophy. Female patients typically develop spinal cord disease and peripheral neuropathy. Predicting the clinical outcome of an individual patient remains impossible due to the lack of genotype-phenotype correlation and predictive biomarkers.
**Methods** The availability of a large prospective cohort of well-characterized patients and associated biobank samples allowed us to investigate the relationship between lipidome and disease severity in ALD. We performed a lipidomic analysis of plasma samples from 24 healthy controls, 92 male and 65 female ALD patients.
**Results** Here we show that VLCFA are incorporated into different lipid classes, including lysophosphatidylcholines, phosphatidylcholines, triglycerides, and sphingomyelins. Our results show a strong association between higher levels of VLCFA-containing lipids and the presence of leukodystrophy, adrenal insufficiency, and severe spinal cord disease in male ALD patients. In female ALD patients, VLCFA-lipid levels correlate with X-inactivation patterns in blood mononuclear cells, and higher levels are associated with more severe disease manifestations. Finally, hematopoietic stem cell transplantation significantly reduces, but does not normalize, plasma C26:0-lysophosphatidylcholine levels in male ALD patients. Our findings are supported by the concordance of C26:0-lysophosphatidylcholine and total VLCFA analysis with the lipidomics results.
**Conclusions** This study reveals the profound impact of ALD on the lipidome and provides potential biomarkers for predicting clinical outcomes in ALD patients.

## Plain language summary

X-linked adrenoleukodystrophy (ALD) affects the brain, spinal cord, and adrenal glands. ALD is caused by too many very long-chain fatty acids (VLCFAs) in the body. We don't know how ALD progresses in individual patients. We have analyzed blood samples from male and female ALD patients. We found that certain changes in fatty acid (or lipid) composition are associated with more severe symptoms. Our findings may lead to new ways to predict which symptoms are likely to change over time and to monitor the effectiveness of treatment. This research increases our understanding of ALD and may improve patient care in the future.

X-linked adrenoleukodystrophy (ALD; OMIM: 300100) results from pathogenic variants in the *ABCD1* gene and affects 1 in 14.700 births[1,2]. The *ABCD1* gene encodes the ALD protein (ALDP), the peroxisomal membrane transporter of very long-chain fatty acids-CoA esters (VLCFA; ≥C22:0) into peroxisomes[3]. ALDP dysfunction impairs peroxisomal β-oxidation, leading to VLCFA accumulation in tissues[4,5]. Disease expression is unpredictable, even in monozygotic twins[6–9], and is influenced by largely unknown genetic and environmental factors[6,10]. Patients are asymptomatic at birth and symptoms accumulate over time. Adrenocortical failure affects 80% of male patients, with half of affected boys becoming symptomatic before the age of 10 years[11]. All males develop spinal cord disease and peripheral neuropathy in adulthood, with highly variable onset (peaking in the 3rd and 4th decades) and progression[12].

In addition, male patients have a 60% lifetime risk of developing a highly characteristic leukodystrophy known as cerebral ALD (CALD), with 30–40% of cases occurring between the ages of 3 and 18 years[12]. Untreated, this leukodystrophy progresses rapidly, often leading to severe disability or death within two years of symptom onset[12]. Hematopoietic stem cell transplantation (HCT) effectively halts the progression of cerebral ALD, but early intervention is required for a favorable outcome[12]. Women with ALD develop spinal cord disease and peripheral neuropathy, typically in their 5th decade, with slower progression than men[13,14]. Leukodystrophy and adrenocortical insufficiency are exceptionally rare in women[13,14].

Newborn screening and subsequent follow-up of boys diagnosed with ALD improves outcomes through timely initiation of treatment[15,16]. However, a test to predict individual risk for adrenal insufficiency or cerebral ALD is lacking. Therefore, all diagnosed boys follow a rigorous protocol that includes adrenal function testing starting at 3 months of age and frequent brain MRI scans starting at age 2 years[12]. To improve follow-up, sensitive biomarkers are needed to monitor and/or predict disease progression.

Previous studies have failed to correlate plasma VLCFA levels with disease severity[17–19]. These studies measured VLCFA as total C22:0, C24:0, and C26:0 fatty acids, which requires a preparatory step to release fatty acids from complex lipids, thereby losing information about the biomolecular origin. Another confounding factor is that individuals included in those studies may have developed new symptoms after the time of assessment. Thus, large prospective cohorts with well-characterized patients and accurate classification criteria are needed to detect correlations between biochemical biomarkers and disease severity. Various studies show that excessive VLCFA in ALD is incorporated into complex lipids such as phosphatidylcholines, cholesterol esters, and triglycerides[20–24] and are likely to drive neurodegeneration. Increased VLCFA incorporation disrupts lamellar lipid profiles, leading to significant morphological perturbations in adrenal striated cells, testicular interstitial cells, Schwann cells, and brain macrophages[25]. It also alters erythrocyte membrane viscosity[26] and disrupts membrane structure and function in artificial phospholipid vesicles[27]. Analysis of postmortem cerebral ALD brain tissue revealed high levels of VLCFA in phosphatidylcholines (PC) in all brain regions[20], especially in cholesterol esters (CE) in actively demyelinating areas. Importantly, VLCFA-rich CE was not observed in normal-appearing white matter. VLCFA-containing lysophosphatidylcholine (LPC) may play a role in cerebral ALD. Injecting LPC(24:0), but not LPC(16:0), into the parietal cortex of wild-type mice, resulted in microglial apoptosis resembling the features of cerebral ALD[28]. Mass spectrometry-based lipidomics is a powerful approach for the comprehensive assessment of the total lipid content, the lipidome, of a biological sample using analytical chemistry. While some studies have applied lipidomics to ALD research, the correlation between lipid profiles of ALD patients and disease manifestation remains unclear due to small sample sizes and the inability to correlate findings with clinical outcomes[29–31].

In this study, we investigate the relationship between lipidome and disease severity in ALD using a large prospective cohort of well-characterized patients and associated biobank samples. Our results show a strong association between higher levels of VLCFA-containing lipids and the presence of cerebral ALD, adrenal insufficiency, and severe spinal cord disease in male ALD patients. In female ALD patients, we find that VLCFA-lipid levels correlate with X-inactivation patterns, and higher levels are associated with more severe disease manifestations. Our findings are supported by the concordance of LPC(26:0) and total VLCFA analysis with lipidomics results. Overall, this study provides robust evidence of a strong association between plasma VLCFA-lipid levels and disease severity in both male and female ALD patients.

## Methods
### Patient selection and clinical assessment
Plasma samples were collected from 24 healthy controls (12 males and 12 females) and 178 ALD patients (112 males and 66 females) from the biobank associated with the "Dutch ALD cohort" at Amsterdam UMC

(130 patients)[32], the German Center of Excellence for ALD at Leipzig UMC (32 patients) and the University of Minnesota Division of Pediatric Blood and Marrow Transplant and Cellular Therapies (16 patients). The study was conducted in accordance with the guidelines of the Declaration of Helsinki. At each participating center, written informed consent was obtained from all subjects enrolled in the study and/or their parents or legal guardians in the case of minors. The protocols for the ongoing prospective cohort studies (including approval to share samples with external investigators) were approved by the local institutional review boards of Amsterdam UMC (METC 2018–310), Leipzig UMC (ek-371/21), and the University of Minnesota (0808M42321). The diagnosis of ALD was confirmed retrospectively by the presence of a hemizygous or heterozygous pathogenic *ABCD1* variant in male and female patients. In this study, "male" or "female" refers to biological sex as determined genetically by the presence of one or two X chromosomes and recorded in the research database for all study participants. Clinical information was available for all patients at the time of blood sampling. Cerebral ALD was defined as leukodystrophy on brain MRI in a pattern consistent with ALD[33]. Adrenal insufficiency was considered to be present in ALD patients receiving hormone replacement therapy, but not when only stress dosing was prescribed. Spinal cord disease was defined as previously described[14,34]. If gait disturbance due to spinal cord disease was present, Expanded Disability Status Scale (EDSS) scores were used to quantify severity[35]. Patients did not receive Lorenzo's oil. Patients did not receive HCT unless specifically indicated. All biochemical analyses were performed at the Amsterdam UMC.

### Patient stratification lipidomics study male ALD patients
In this study, our main objective was to find lipids that correlate with disease severity. To achieve this, we categorized 92 male ALD patients based on the presence of cerebral ALD, adrenal insufficiency, and severe spinal cord disease (EDSS > 6) at the time of assessment (Table 1). As with all progressive disorders, a confounding factor is that individuals may develop new symptoms after the time of assessment. To mitigate this effect, we defined categories using knowledge of natural history[11,32,36]. For example, if a male patient has normal adrenal function at age 55 or later, it is likely to remain normal[11]. Similarly, leukodystrophy rarely occurs after the age of about 55 years. Severity of spinal cord disease is associated with the age of onset; those with an EDSS score ≤ 6 in late adulthood are likely to remain ambulatory. Therefore, age (over 55 years) was used in addition to symptoms for final classification. Table 1 provides an overview of the different patient categories used in the lipidomics analysis.

### Patient stratification lipidomics study female ALD patients
Lipidomics was performed on plasma samples from 65 female ALD patients. Similar to the approach with male ALD patients, we stratified female ALD patients based on the presence and severity of symptoms. Spinal cord disease is rare before the 4th decade of life[13,14], and therefore we stratified women over the age of 40 ($n = 54$) into two distinct groups based on the presence of spinal cord disease, referred to as "mild" ($n = 26$) and "severe" ($n = 28$). Clinical assessment was performed as previously described[13]. Of all female patients, one had cerebral ALD and none had evidence of adrenal insufficiency.

### Lipidomics analysis
Lipidomics analysis was performed as previously described[14,37]. Briefly, 20 μL of plasma was added to a 2 mL tube and spiked with internal standards for different lipid classes. These included 0.5 nmol lysophosphatidylcholine LPC(14:0), 2.0 nmol phosphatidylcholine PC(14:0/14:0), 0.02 nmol lysophosphatidylglycerol LPG(14:0), 0.1 nmol phosphatidylglycerol PG(14:0/14:0), 0.5 nmol diglycerides DG(14:0/14:0), 0.5 nmol triglycerides TG(14:0/14:0/14:0), 0.1 nmol lysophosphatidic acid LPA(14:0), 0.5 nmol phosphatidic acid PA(14:0/14:0), 2.125 nmol sphingomyelin SM(d18:1/12:0), 0.1 nmol cardiolipin CL(14:0/14:0/14:0/14:0), 0.1 nmol lysophosphatidylethanolamine LPE(14:0), 0.5 nmol phosphatidylethanolamine PE(14:0/14:0), 0.5 nmol phosphatidylinositol PI(8:0/8:0), 5.0 nmol

**Table 1 | Overview patient stratification lipidomics**

| | Groups | n | Mean age and range (years) | Adrenal insufficiency | Cerebral ALD | Mean EDSS and range |
|---|---|---|---|---|---|---|
| Presence of Cerebral ALD (males) | CALD | 24 | 38 (6−73) | 19 (79%) | 24 (100%) | 3.6 (0−9) |
| | NoCALD (all ages) | 68 | 40 (9−74) | 31 (45%) | 0 | 3 (0−7) |
| | NoCALD >55 | 21 | 65 (56−74) | 3 (14%) | 0 | 5 (1.5−7) |
| Presence of adrenal insufficiency (males) | AI | 50 | 33 (6−73) | 50 (100%) | 19 (38%) | 2.4 (0−7) |
| | NoAI (all ages) | 42 | 50 (9−74) | 0 | 5 (12%) | 4.1 (0−9) |
| | NoAI >55 | 20 | 65 (56−74) | 0 | 2 (10%) | 5.2 (1.5−7.5) |
| Severity of Spinal cord disease (males) | Mild >55 | 17 | 65 (56−74) | 1 (6%) | 1 (6%) | 4.6 (1.5−6) |
| | Severe | 15 | 54 (23−73) | 7 (47%) | 5 (33%) | 6.8 (6.5−9) |
| Severity of Spinal cord disease (females) | Mild | 28 | 58 (41−78) | 0 | 0 | 2.4 (0−6) |
| | Severe | 26 | 59 (41−74) | 0 | 1 (3%) | 4 (2−6.5) |

phosphatidylserine PS(14:0/14:0), 2.5 nmol cholesterol ester D7-CE(16:0), and 0.125 nmol sphingosine and ceramide mixture (Avanti Polar Lipids), all dissolved in 1:1 (v/v) methanol:chloroform. To each sample, 1.5 mL of 1:1 (v/v) methanol:chloroform was added, followed by 5 min of sonication in a water bath. The samples were then centrifuged at $16,000 \times g$ for 10 min at 4 °C. The supernatant was transferred to a 1.5 mL glass autosampler vial and evaporated under a nitrogen stream at 45 °C. Finally, the dried lipids were reconstituted in 100 µL of 1:1 (v/v) chloroform:methanol. Lipid chromatographic separation was performed on a Thermo Fisher Scientific Ultimate 3000 binary UPLC system, using both normal-phase and reverse-phase columns in separate runs. The normal-phase separation was performed on a Phenomenex® LUNA silica column (250 × 2 mm, 5 µm, 100 Å) with the column temperature maintained at a constant 25 °C. Mobile phase A for the normal-phase separation consisted of 85:15 (v/v) methanol:water containing 0.0125% formic acid and 3.35 mmol/L ammonia, while mobile phase B consisted of 97:3 (v/v) chloroform:methanol containing 0.0125% formic acid. The LC gradient was programmed as follows 10% A from 0–1 min, increasing to 20% A at 4 min, then to 85% A at 12 min, followed by 100% A at 12.1 min, held at 100% A from 12.1–14 min, returning to 10% A at 14.1 min, and maintained at 10% A from 14.1–15 min. The flow rate was set at 0.3 mL/min. A Waters HSS T3 column (150 × 2.1 mm, 1.8 µm particle size) was used for the reverse-phase separation. The mobile phase A was 4:6 (v/v) methanol:water, and the mobile phase B was 1:9 (v/v) methanol:isopropanol, both containing 0.1% formic acid and 10 mmol/L ammonia. The gradient started at 100% A, transitioning to 80% A at 1 min, then to 0% A at 16 min, held at 0% A from 16–20 min, returning to 100% A at 20.1 min, and held at 100% A until 21 min. The column temperature was maintained at 60 °C with a flow rate of 0.4 mL/min. Lipid detection was performed using a Q Exactive Plus Orbitrap mass spectrometer (Thermo Scientific) in both negative and positive ionization modes. The spray voltage was set at 2500 V with nitrogen as the nebulizing gas. A resolution of 280,000 was applied over a mass range of m/z 150 to m/z 2000.

**Quantification of LPC(26:0)**
Quantification of LPC(26:0) was performed as described earlier[38]. Briefly, 10 µL plasma was extracted with 10 µL of an internal standard solution containing 1 µmol/L D4-C26:0-lysoPC in 0.5 mL of acetonitrile by ultrasonication for 5 min in a sonicator bath (Branson 3510) at room temperature. After centrifugation (5 min, $16,000 \times g$) the resulting methanol (DBS) or acetonitrile (plasma) layer was transferred to a new glass tube and evaporated under a constant stream of nitrogen at 40 °C. The samples were then reconstituted in 50 µL methanol, transferred to a sample vial, and capped. Samples were injected using an ACQUITY UPLC system (Waters, Milford, MA, USA) on a 50 × 2.1 mm, 2.6 µm particle diameter Kinetex C8 column (Phenomenex, Torrance, CA, USA). The column was held at a constant temperature of 50 °C. The composition of mobile phase A was 0.1% formic acid in water and mobile phase B was 0.1% formic acid in

methanol. The gradient used was as follows: T = 0 min: 36% A, 64% B, flow 0.4 mL/min towards T = 6 min: 0% A, 100% B, flow 0.4 mL/min; T = 6–11 min: 0% A, 100% B, flow 0.4 mL/min, and T = 11–11.1 back to 36% A, 64% B, flow 0.4 mL/min. Detection was done using a Quattro Premier XE (Waters, Milford, MA, USA) using electrospray ionization in positive mode. The source temperature was 130 °C, and the capillary voltage was 3.5 kV. Multiple reaction monitoring (MRM) was done on masses 636.50 > 104.10 and 640.50 > 104.10 with a dwell time of 0.03 s. Argon was used as a collision gas.

**Total VLCFA analysis**
Total VLCFA quantification in plasma was performed as previously described[39]. Briefly, 100 µL of plasma was combined with 100 µL of an internal standard solution containing $^2H_4$-labeled fatty acids (C22:0, C24:0, and C26:0) in toluene, and 1 mL of acetonitrile/37% hydrochloric acid (4:1, v/v) in a 4 mL glass vial. The vial was sealed and incubated at 90 °C for 2 h to hydrolyze the samples. After cooling, free VLCFA were extracted with 2 mL of hexane, vortexed, and centrifuged. Approximately 75% of the upper phase was collected, and hexane was evaporated under nitrogen. The fatty acids were dissolved in chloroform-methanol-water (50:45:5, v/v/v) with 0.01% aqueous ammonia and transferred to autosampler vials. The samples were analyzed using a Quattro II triple quadrupole mass spectrometer in negative ESI mode. The system included an HP1100 series binary gradient pump and a Gilson 231 XL autosampler. Nitrogen was used as the nebulizer gas, and the source temperature was 80 °C with a capillary voltage of 3 kV. A 5 µL sample was injected directly into the mass spectrometer, with a flow rate starting at 20 µL/min, increasing to 50 µL/min. Data were acquired with MassLynx NT software, and specific m/z values for endogenous and labeled fatty acids were monitored.

**Determination of X-inactivation**
X-inactivation in peripheral blood mononuclear cells (PBMCs) was determined using RNA-sequencing data using the R package XCIR as described by Sauteraud et al.[40]. Blood was collected in PAXgene Blood RNA Tubes (Qiagen) and processed according to manufacturer recommendations prior to storage at −80 °C. RNA sequencing was done as described earlier[41]. RNA-sequencing libraries were prepared from 200ng total RNA using the KAPA mRNA Hyperprep kit (KAPA biosystems). Libraries were sequenced using the Novaseq 6000 instrument (Illumina) using a Novaseq S4 Flow Cell PE150. The sequencing depth was approximately 40 million reads per sample.

**Bioinformatics, statistics, and reproducibility**
The raw LC/MS data were converted to mzXML format using MSConvert[42]. Lipidomics data processing and analysis were done as described earlier[37]. Briefly, lipidomics data were analyzed using an in-house developed lipidomics pipeline based on the R programming

language (http://ww.r-project.org) and MATLAB. Preprocessing was done using the R package XCMS with minor changes to some functions to better suit the Q Exactive™ data; notably, the definition of noise level in centWave was adjusted, and the stepsize in fillPeaks[43]. Lipid identification was performed using a combination of accurate mass measurements, relative retention times, injection of relevant standards, and analysis of samples with known metabolic defects. Lipid classes in our lipidomics pipeline were defined by their generic chemical formulas, where 'R' denotes the radyl group. Upon importing the lipid database into the annotation pipeline, the generic chemical formula for each lipid class was expanded by replacing the 'R' element with a range of potential radyl group lengths and degrees of unsaturation. This expanded list of chemical formulas was then used to calculate the neutral monoisotopic mass of each species. The monoisotopic mass was then converted into a set of m/z values corresponding to each adduct and charge combination, allowing reliable measurement and annotation of the lipid species. The reported lipid abundances are semi-quantitative, calculated by dividing the analyte response (peak area) by that of the corresponding internal standard, with the result further adjusted for the concentration of the internal standard, and expressed in arbitrary units (A.U.). Statistical analyses were conducted utilizing R version 4.0.2 and Prism version 9.5.1. The normality of the data distribution was assessed using a Shapiro–Wilk test. Depending on the distribution, either a Welch's $t$-test or a Mann–Whitney $U$ test was employed to compare groups. For comparisons across multiple groups, a Kruskal–Wallis test followed by Dunn's post hoc tests was used. To correct for multiple testing, a post hoc Benjamini–Hochberg adjustment was applied to the $p$-values on a per-comparison basis. Significance was established with a threshold of $p < 0.05$ after Benjamini–Hochberg adjustment. In addition, Bonferroni and Holm adjusted $p$-values are provided in Supplementary Data 2 and 4. Correlation between continuous variables was investigated using the two-tails Spearman correlation test. Lipid saturation and chain length plots were generated using the R package lipidr version 2.15.1[44]. In lipidr, chain length plots are created by plotting a regression line (LOESS curve) of the (log2) fold changes and the total chain lengths of lipids within a specific lipid class.

### Reporting summary

Further information on research design is available in the Nature Portfolio Reporting Summary linked to this article.

## Results

### ALD results in VLCFA incorporation in complex lipids

Lipidomics analysis of plasma samples from 92 male ALD patients and 12 healthy male controls identified 1556 lipids (Supplementary Data 1) and revealed that the lipidome of ALD patients was significantly different from that of controls, as indicated by the results of principal component analysis (PCA) and partial least squares discriminant analysis (PLS-DA) (Fig. 1A). In ALD patients, 421 lipids were significantly elevated (>1.5-fold change, $p < 0.05$), and 45 lipids were significantly lower after false discovery rate correction (Fig. 1B, Supplementary Data 2). Elevated lipids included species from multiple lipid classes such as LPC, PC, diacylglycerol (DG), triglyceride (TG), CE, and others (Fig. 1C). One of the most elevated lipids was the 1-acyl form of LPC(26:0), which is the marker in ALD newborn screening[45]. As expected, 1-acyl LPC(26:0) showed a clear separation between ALD patients and controls with no overlap ($p = 0.0000007$; Fig. 1C). Total acyl-chain lengths of the elevated lipids in ALD patients were characteristic of VLCFA incorporation (Fig. 1D). For single acyl-chain containing lipids (like LPC and CE), the total acyl-chain length was >20 carbons with a maximum of two double bonds (Fig. 1D, E). For two acyl chains containing lipids (like PC and DG), the total acyl-chain length was >40 carbons with up to 7 double bonds. For TG, which contain three acyl chains, the total acyl-chain length exceeded 60 carbons with up to 7 double bonds. Interestingly, lower levels of shorter polyunsaturated TG were observed in ALD patients. Furthermore, the results showed a correlation between lipid fold increase and total acyl-chain length. In ALD patients, greater total acyl-chain length corresponded

to a larger fold increase, demonstrating that VLCFA were incorporated into various lipid classes, substantially altering the lipidome.

### Cerebral ALD in males is associated with higher levels of VLCFA-containing lipids

To explore the relationship between the lipidome and the presence of cerebral ALD, we compared patients with cerebral ALD (CALD) to patients without cerebral ALD (NoCALD). The lipidome of the CALD group showed several elevated lipids compared to the NoCALD group (Fig. 2A, B). The most notable differences were observed in lipids within the LPC, PC, and TG classes. These lipids had total acyl-chain lengths that were characteristic of VLCFA incorporation (Fig. 2C, D). Elevated LPC had a total acyl-chain length of greater than 20 carbons with zero, one, or two double bonds. Elevated PC had a total acyl-chain length greater than 40 carbons, with up to six double bonds. In addition, the increased levels of TG showed a total acyl-chain length of over 60 carbons with up to four total double bonds. Notably, the fold increase in CALD compared to NoCALD indicated a trend with increasing total acyl-chain length, as shown in Fig. 2C for LPC, PC, and TG. 1-acyl LPC(32:0) showed the most significant statistical difference in CALD compared to NoCALD ($p = 0.00056$). In addition, CALD was associated with lower levels of shorter lipids for single acyl-chain lipids, two acyl-chain lipids, and TG compared to NoCALD (Fig. 2D). The differences in elevated lipid levels between CALD and NoCALD became more pronounced when comparing patients without cerebral ALD aged >55 years (Fig. 2B). As expected, compared to controls, patients without cerebral ALD aged >55 years still have a lipidome profile characteristic of ALD (Supplementary Fig. 1).

Samples from three patients before and after the onset of cerebral ALD were available and included in the analysis. As shown by the dashed lines in Fig. 2B, all three patients had VLCFA-containing lipid levels within the top 50% of patients without apparent cerebral involvement before the onset of CALD. Furthermore, VLCFA-containing lipid levels remained relatively stable or increased after the onset of CALD. These results may therefore suggest that ALD patients without cerebral involvement and high levels of VLCFA-containing lipids may be particularly susceptible to developing CALD. Further studies in large cohorts of patients with samples taken before and after the onset of cerebral ALD are essential to better understand how VLCFA-containing lipid levels influence the risk of developing CALD.

### Adrenal insufficiency is associated with higher levels of VLCFA-containing lipids

To investigate the potential association between the lipidome and adrenal insufficiency (AI), we compared lipid profiles in patients with and without AI (NoAI). Our results showed significantly elevated levels of VLCFA-containing lipids in several lipid classes, including LPC, PC, and TG, in the AI group compared with the NoAI group (Fig. 3A, B). The elevated lipids exhibited characteristic acyl-chain lengths indicative of VLCFA incorporation (Fig. 3C, D). Conversely, lipids with shorter acyl-chain lengths showed minimal differences or slightly lower levels in the AI group, whereas lipids with the longest total acyl-chain lengths showed the most significant differences. The difference in VLCFA-lipid levels became more pronounced when patients without adrenal insufficiency over the age of 55 years were selected. As expected, compared to controls, patients without AI aged >55 years still have a lipidome profile typical of ALD (Supplementary Fig. 1). These results demonstrate that higher VLCFA-lipid levels are associated with the presence of AI.

### Severity of spinal cord disease in male ALD patients is correlated with VLCFA-lipid levels

Spinal cord disease in ALD patients is progressive throughout life, with a considerable variability in the age of onset. This variability poses a significant challenge for patient stratification, as some patients may be asymptomatic at the time of sampling but remain at risk for developing symptoms later in life. To address this issue, we compared the lipid profile in patients with severe (EDSS > 6) versus mild (EDSS ≤ 6) spinal cord disease aged >55 years.

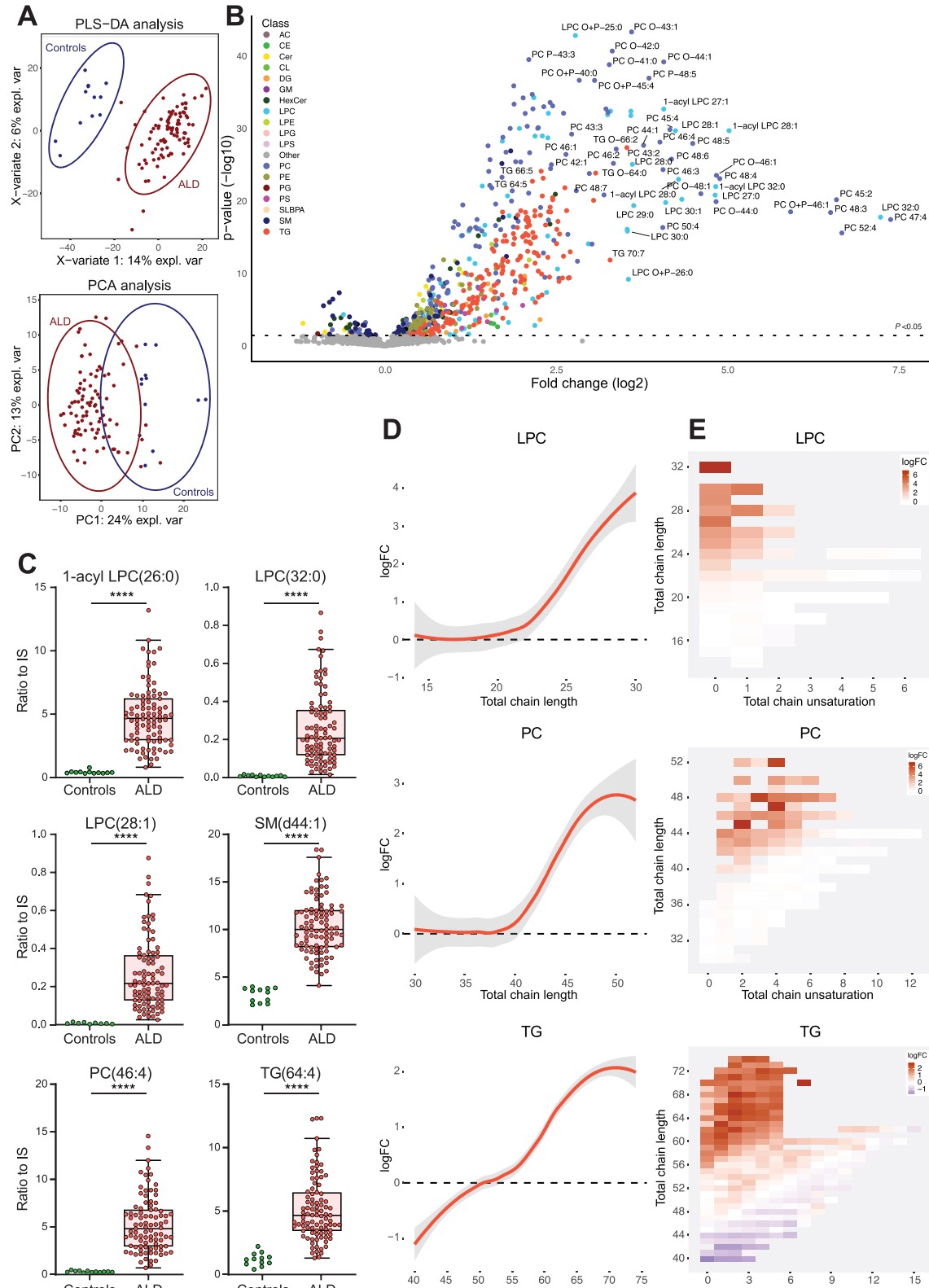

**Fig. 1 | Lipidomics analysis of male ALD patients versus controls.** Lipidomics analysis of plasma samples from 92 male ALD patients and 12 healthy male controls. **A** Partial least squares discriminant analysis (PLS-DA) and principal component analysis (PCA) analysis. **B** Volcano plot of lipid levels. The vertical axis shows the *p*-value (−log10) from Welch's *t*-tests between ALD and controls, and the horizontal axis shows the fold change (log2) between ALD and controls. Colored dots are lipids with a *p*-value < 0.05. **C** Box plots (whiskers are determined using Tukey) of elevated lipids in ALD compared to controls. **D** Trend lines showing log fold change (logFC) and total chain length of lysophosphatidylcholine (LPC), phosphatidylcholine (PC), and triglyceride (TG) lipids in ALD compared to controls. **E** Heatmap showing log fold change, total chain unsaturation, and chain length for LPC, PC, and TG. Welch's *t*-test or Mann–Whitney *U* test was used to determine significant differences between groups (****$P \leq 0.0001$).

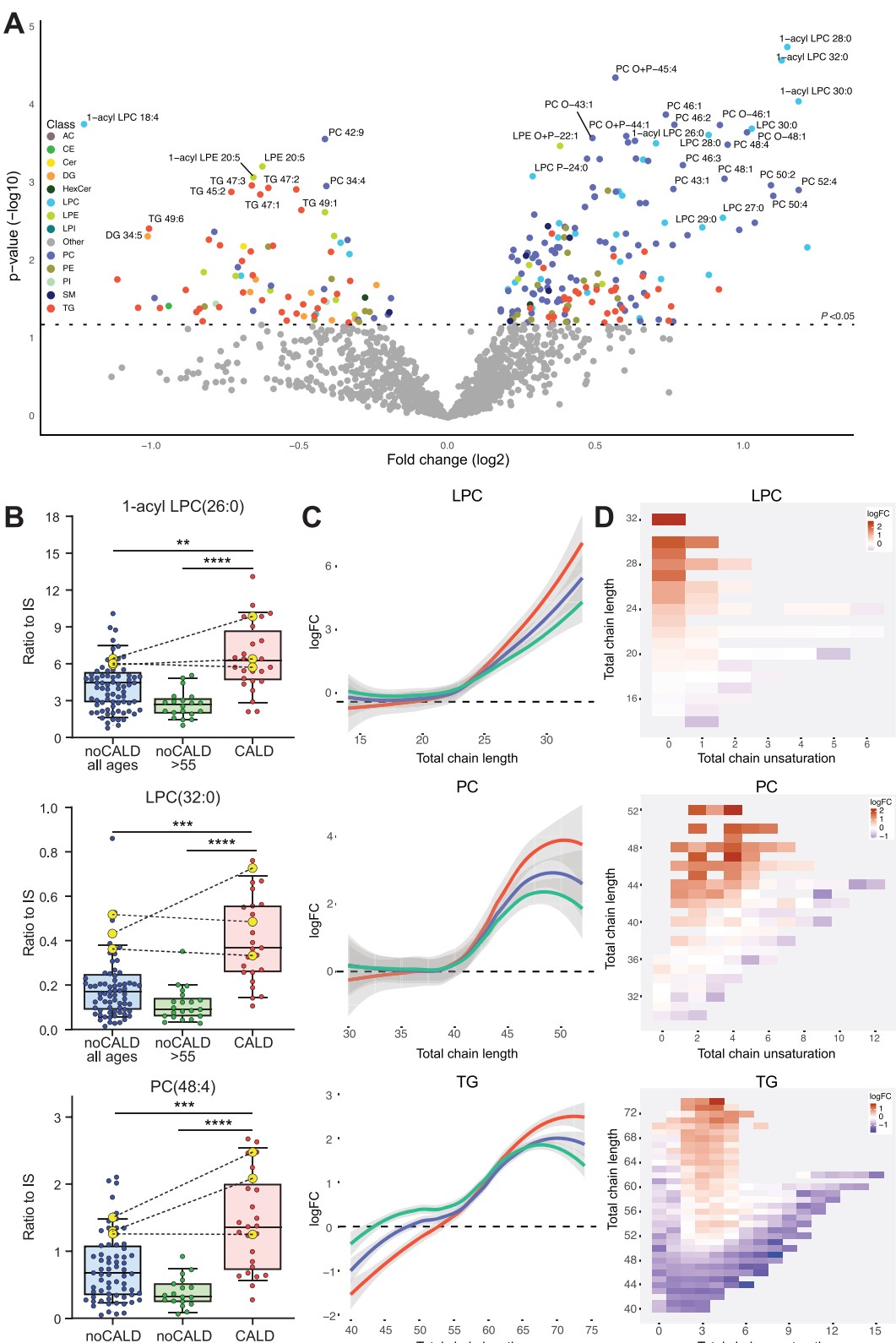

**Fig. 2 | Correlation of the lipidome with the presence of cerebral ALD.** Lipidomics analysis of plasma samples from patients with cerebral ALD (CALD) (*n* = 24) and patients without CALD (noCALD) (all ages, *n* = 68). **A** Volcano plot of lipid levels. The vertical axis shows the *p*-value (−log10) from Welch's *t*-tests between CALD (*n* = 24) and noCALD (all ages, *n* = 68), and the horizontal axis the fold change (log2) between CALD and noCALD (all ages). Colored dots are lipids with a *p*-value < 0.05. **B** Box plots (whiskers are determined using Tukey) of elevated lipid levels in CALD compared to noCALD (all ages), and noCALD >55 years (*n* = 21) are shown. Yellow dots on the graph represent patients who were sampled both before and after the onset of CALD (*n* = 3). **C** Trend lines showing log fold change (logFC) and total chain length of lysophosphatidylchloline (LPC), phosphatidylcholine (PC), and triglyceride (TG) lipids comparing patient groups with controls (CALD: red, noCALD all ages: blue, noCALD >55: green). **D** Heatmap showing the log fold change, total chain unsaturation, and chain length for LPC, PC, and TG when CALD is compared to noCALD (all ages). Welch's *t*-test or Mann–Whitney *U* test was used to determine significant differences between groups (**P ≤ 0.01; ***P ≤ 0.001), (****P ≤ 0.0001).

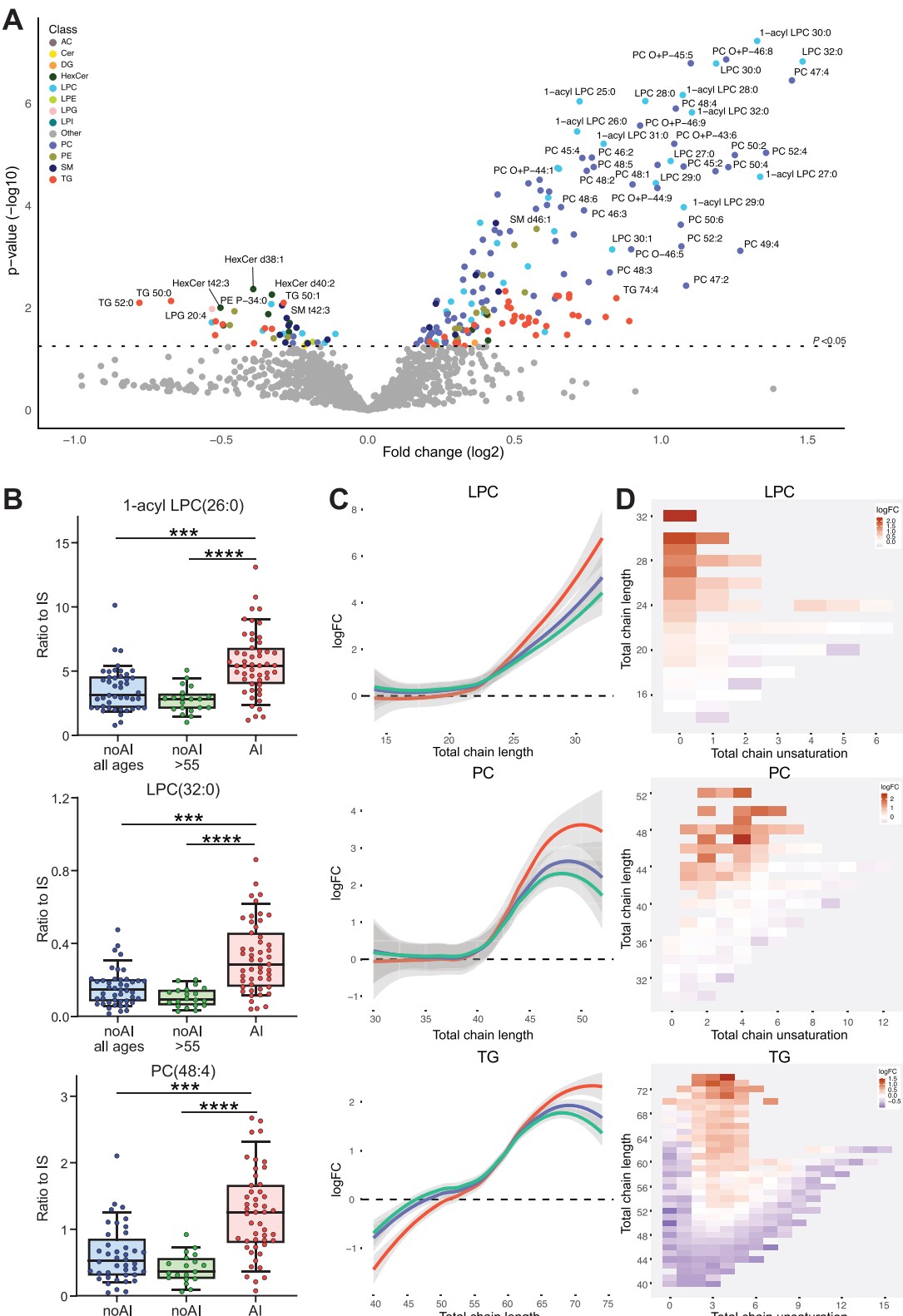

**Fig. 3 | Correlation of the lipidome with the presence of adrenal insufficiency.** Lipidomics analysis of plasma samples from patients with adrenal insufficiency (AI) (n = 50) and patients without AI (NoAI) (all ages, n = 42). **A** Volcano plot of lipid levels. The vertical axis shows the p-value (−log10) from Welch's t-tests between AI and NoAI, and the horizontal axis shows the fold change (log2) between AI and noAI (all ages). Colored dots are lipids with a p-value of <0.05. **B** Box plots (whiskers are determined using Tukey) of elevated lipid levels in AI compared to noAI (all ages), and noAI >55 years (n = 20) are shown. **C** Trend lines showing log fold change (logFC) and total chain length of LPC, PC, and TG lipids when comparing patient groups to controls (AI: red, noAI all ages: blue, noAI >55: green). **D** Heatmap showing log fold change, total chain unsaturation, and chain length for lysophosphatidylcholine (LPC), phosphatidylcholine (PC), and triglyceride (TG) when AI is compared to noAI (all ages). Welch's t-test or Mann–Whitney U test was used to determine significant differences between groups (***P ≤ 0.001), (****P ≤ 0.0001).

Our data show that the severe group had higher levels of VLCFA-lipids, mainly from the LPC and PC classes (Fig. 4A, B). No significant differences in TG levels were observed between the two groups. Compared to controls, patients with mild spinal cord disease still have a lipidome profile characteristic of ALD (Supplementary Fig. 1).

## Disease severity in female ALD patients is correlated with VLCFA-lipid levels

We performed lipidomics analysis on the plasma of 65 female ALD patients and compared the profile with that of 12 healthy female controls (Supplementary Data 3). PCA and PLS-DA revealed that ALD affects the lipidome of female patients (Fig. 5A). Compared to controls, female patients had elevated levels of VLCFA-containing lipids, including LPC, PC, CE, and TG (Fig. 5B, Supplementary Data 4). Similar to male ALD patients, the severity of spinal cord disease can vary widely among female ALD patients. Lipidomics analysis revealed several moderately elevated lipids associated with the severe group compared to the mild group, including VLCFA-containing species from the LPC, PC, and TG classes (Fig. 5C). In addition, the levels of VLCFA-containing lipids in the severe group increased with longer total acyl-chain lengths (Fig. 5D, E). This trend was true for LPC, PC, and TG. In our lipidomics analysis, we had one female patient with cerebral ALD who consistently had the highest VLCFA-containing lipid levels (Fig. 5C). These results suggest that severe disease in female patients is associated with higher levels of VLCFA-lipids.

## X-inactivation pattern correlates with LPC(26:0) levels in female ALD patients

In females, X-inactivation significantly affects cellular function and gene expression. It ensures balanced expression of X-linked genes despite the presence of two X chromosomes in each cell. Because ALD is an X-linked disorder, X-inactivation patterns have been studied, suggesting a possible association with disease severity and neurological symptoms[46–48]. We hypothesized a correlation between VLCFA-lipid levels in female patients and the *ABCD1* allele-specific expression pattern. Therefore, we measured X-inactivation patterns in peripheral blood mononuclear cells from 28 female patients. Our results showed a strong linear correlation between plasma LPC(26:0) levels and X-inactivation patterns with a Pearson correlation coefficient of 0.79 (Fig. 5F). Preferential expression of the *ABCD1* variant allele was associated with higher LPC(26:0) levels. Notably, the ALD female with cerebral ALD showed an X-inactivation pattern completely skewed toward the allele harboring the pathogenic variant. Taken together, these findings demonstrate a strong association between X-inactivation and LPC(26:0) levels.

## Plasma LPC(26:0) concentration determined by targeted assay correlates with disease severity in ALD patients

LPC(26:0) analysis as determined by UPLC-MS/MS plays a key role in ALD diagnosis, ALD newborn screening, and the assessment of VLCFA accumulation in other peroxisomal disorders[38]. Compared to lipidomics, this method provides more accurate concentration values in a sample, allowing for better longitudinal comparison. We expanded the cohort to include an additional 20 male and 1 female patients and performed targeted LPC(26:0) analysis on plasma samples from the 112 male and 66 female patients included in this study, as shown in Fig. 6A and Supplementary Data 5. The targeted LPC(26:0) results closely parallel the lipidomics results. Higher concentrations of LPC(26:0) were associated with the presence of cerebral ALD, adrenal insufficiency, and severe spinal cord disease (Fig. 6A, Supplementary Data 6). In addition, Supplementary Fig. 2 illustrates LPC(26:0) levels in patient groups categorized by the presence or absence of cerebral ALD and/or adrenal insufficiency (Supplementary Fig. 2A) and spinal cord severity (Supplementary Fig. 2B). The highest LPC(26:0) levels were observed in patients with cerebral ALD and adrenal insufficiency. Conversely, patients without one or both of these conditions had lower LPC(26:0) levels. In female ALD patients, higher LPC(26:0) levels were associated with more severe disease. In addition, we obtained and included

an additional plasma sample from a second female ALD patient with cerebral ALD. In these two female CALD samples, LPC(26:0) levels were similar to those observed in male patients with cerebral ALD (Fig. 6A). Table 2 summarizes the observed concentration ranges for the different patient groups. We also examined LPC(26:0) concentrations in 1090 healthy controls (both male and female) aged 0 to 78 years. Our results show no sex or age-related effects (Fig. 6B). Taken together, these findings align targeted LPC(26:0) analysis with lipidomics and emphasize that higher LPC(26:0) concentrations are associated with more severe disease presentation in both male and female ALD patients.

## Plasma LPC(26:0) levels decrease post HCT

Currently, HCT is the preferred therapy for cerebral ALD. To evaluate its effect on LPC(26:0) levels, we collected multiple samples from 12 male CALD patients who underwent HCT at baseline and at various time points after the procedure. Before HCT, patients had a mean LPC(26:0) concentration of 435 nmol/L (range: 228–576 nmol/L) (Fig. 6C). After HCT, we observed a substantial and statistically significant decrease in LPC(26:0) concentration over time. At 12 months after HCT, the mean LPC(26:0) concentration decreased to 219 nmol/L (range: 137–334 nmol/L). In addition, plasma samples from 5 patients at 24 months post HCT showed a mean LPC(26:0) concentration of 190 nmol/L (range: 106–267 nmol/L). Despite these reductions, levels remained elevated compared to healthy controls (upper limit of reference range: 72 nmol/L).

## Comparison of plasma VLCFA levels, LPC(26:0) and disease severity

Previous studies have failed to show an association between disease severity and plasma total VLCFA levels, possibly due to smaller non-prospective cohorts with incomplete clinical information[17–19]. Our approach, using defined patient groups avoided the "dilution" of differences. Our findings of associations between plasma VLCFA-containing lipids and disease severity prompted us to investigate whether these associations extended to total VLCFA levels. We analyzed total VLCFA levels in 72 plasma samples from male ALD patients and 45 from female ALD patients (Supplementary Data 7). Of note, we found a significant correlation between total C26:0 levels and disease severity. Specifically, higher C26:0 levels were associated with the presence of cerebral ALD, adrenal insufficiency, and severe spinal cord disease in male patients (Fig. 6D, Supplementary Data 8). Similarly, in female ALD patients higher C26:0 levels were associated with more severe disease. Notably, C26:0 correlated strongly with plasma LPC(26:0) levels, with a Spearman rank correlation coefficient of 0.7 (Fig. 6E). However, it is important to emphasize that LPC(26:0) is the preferred marker in the majority of cases. In contrast to total C26:0, LPC(26:0) is less likely to be influenced by diet, and LPC(26:0) analysis is less labor-intensive compared to C26:0 analysis. In addition, targeted analysis of LPC(26:0) revealed more substantial differences between patient groups and resulted in a better separation, i.e., a greater fold increase compared to C26:0 (as shown in Table 3).

## Discussion

ALD is a monogenic metabolic disorder with a broad clinical spectrum in both male and female ALD patients. Due to the lack of genotype-phenotype correlation and/or predictive biomarkers, it remains impossible to predict the clinical outcome of an individual patient. Newborn screening and the widespread use of next-generation sequencing for routine diagnostics are identifying more presymptomatic patients, underscoring the need for accurate prognostic biomarkers to enable better counseling and personalized follow-up[49]. In this study, we characterized the plasma lipidome of 24 healthy controls (12 males and 12 females) and 157 ALD patients (92 males and 65 females). The aim of the study was to identify potential biomarkers that correlate with ALD disease expression and severity. We categorized male patients based on the presence of cerebral ALD, adrenal insufficiency, and spinal cord disease severity. Stratifying patients by symptoms is challenging due to the progressive nature of ALD, and the absence of symptoms

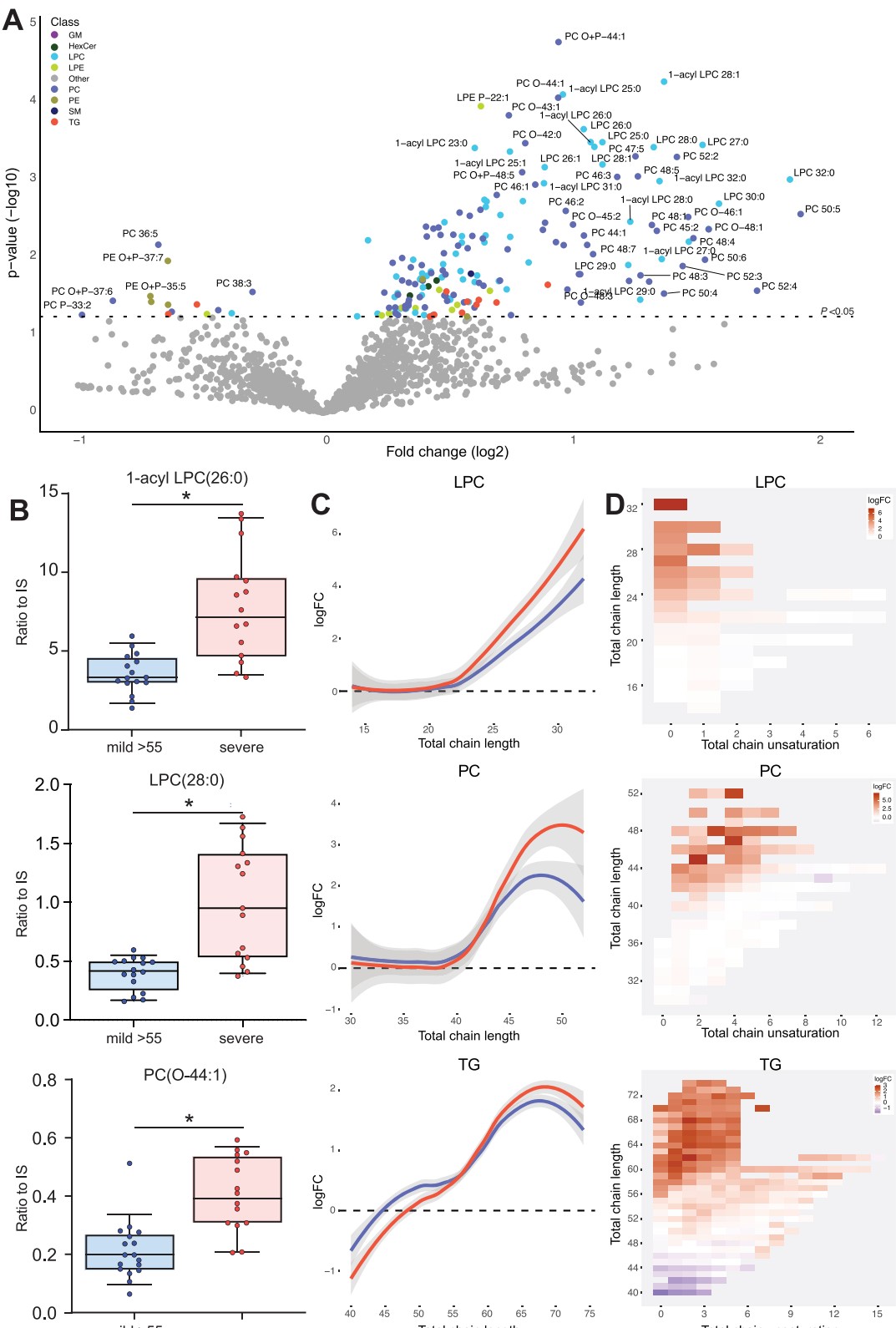

**Fig. 4 | Correlation of the lipidome with the presence of severe spinal cord disease.** Lipidomics analysis of plasma samples from patients with mild (EDSS≤6) spinal cord disease aged >55 years (n = 17) and severe (EDSS>6) spinal cord disease (n = 15) patients. **A** Volcano plot of lipid levels. The vertical axis shows the p-value (−log10) from Welch's t-tests between mild and severe patients, and the horizontal axis shows the fold change (log2) between mild and severe patients. Colored dots are lipids with a p-value of <0.05. **B** Box plots (whiskers are determined using Tukey) of elevated lipid levels in severe compared to mild are illustrated. **C** Trend lines showing log fold change (logFC) and total chain length of lysophosphatidylcholine (LPC), phosphatidylcholine (PC), and triglyceride (TG) lipids when comparing patient groups to controls (severe: red, mild: green). **D** Heatmap showing log fold change, total chain unsaturation, and chain length for LPC, PC, and TG when severe is compared to mild. Welch's t-test or Mann–Whitney U test was used to determine significant differences between groups (*P ≤ 0.05).

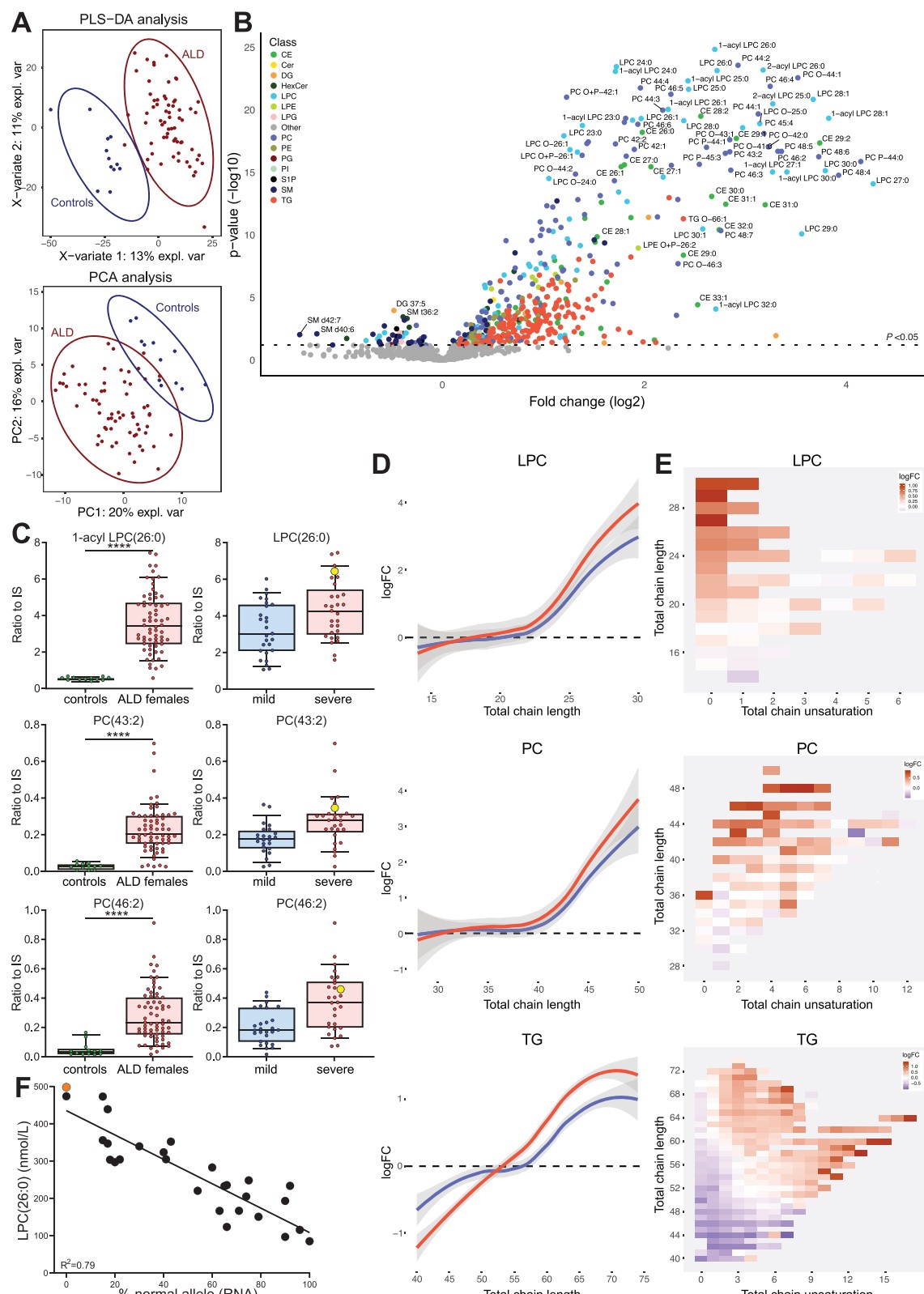

at the time of assessment does not preclude occurrence later in life. However, the likelihood of developing cerebral ALD and adrenal insufficiency decreases with age[11]. Therefore, we also stratified patients by age.

Our results show that the lipidome of both male and female ALD patients has several elevated VLCFA-containing lipids compared to healthy controls. One of the most significantly elevated lipids in ALD was LPC(26:0), which is the marker used for ALD newborn screening. In addition, several other LPC species and species from VLCFA-enriched lipid classes such as PC, TG, and CE are significantly elevated in ALD patients. These findings demonstrate that VLCFA incorporation extends beyond a single lipid class and likely affects various lipid-mediated physiological processes. The significant changes observed in the ALD

**Fig. 5 | Lipidomics analysis of female ALD patients versus controls.** Lipidomics analysis of plasma samples from 65 female ALD patients and 12 healthy female controls. **A** Partial least squares discriminant analysis (PLS-DA) and principal component analysis (PCA) analysis. **B** Volcano plot of lipid levels. The vertical axis shows the $p$-value ($-\log10$) from Welch's $t$-tests between female ALD patients and controls, and the horizontal axis shows the fold change ($\log2$) between female ALD patients and controls. Colored dots are lipids with a $p$-value < 0.05. **C** Box plots (whiskers are determined using Tukey) of lipids that are elevated in female ALD patients compared to controls and are associated with spinal cord disease severity. **D** Trend lines illustrating the log fold change (logFC) and total chain length of lysophosphatidylcholine (LPC), phosphatidylcholine (PC), and triglyceride (TG) lipids when patient groups are compared to controls (severe: red, mild: blue). **E** Heatmap showing the log fold change, total chain unsaturation, and chain length for LPC, PC, and TG when the severe group was compared to the mild group. **F** Linear correlation between plasma LPC(26:0) levels and the X-inactivation in female ALD patients ($n = 28$, Pearson correlation coefficient: 0.79). Orange dot represents the female patient with cerebral ALD. Welch's $t$-test or Mann–Whitney $U$ test was used to determine significant differences between groups (\*\*$P \le 0.01$; \*\*\*\*$P \le 0.0001$).

lipidome highlight the importance of lipidomics in identifying new biomarker candidates.

We sought to identify biomarkers associated with a (higher) risk of cerebral ALD. To this end, we performed a comparative analysis of the plasma lipidome in ALD patients with and without cerebral involvement. The results showed a significant increase in several VLCFA-containing lipids. The most significant differences were observed in LPC, PC, and TG lipids, with the highest levels in patients with cerebral ALD. In addition, there was a decrease in shorter and polyunsaturated lipid variants, particularly in TG lipids, suggesting that the elevated levels of VLCFA-lipids in cerebral ALD patients may be due to alterations in fatty acid homeostasis. When comparing patients without cerebral ALD over the age of 55 with patients with cerebral ALD, the difference in VLCFA-lipid levels is even more pronounced. This increased differentiation underscores the importance of age stratification. Furthermore, analysis of the plasma lipidome of three patients for whom samples were available before the onset of cerebral ALD showed that VLCFA-lipid levels were already high before the onset of cerebral ALD and that the levels remained constant after the onset of cerebral ALD. Taken together, our results suggest that plasma VLCFA-containing lipids may be indicative of the risk of developing cerebral ALD. However, due to the diversity of VLCFA-lipid levels observed, accurate predictions at the individual level remain challenging at present.

VLCFA and VLCFA-containing lipid levels in ALD are influenced by several metabolic processes. These include the availability of fatty acids for VLCFA synthesis, the rate of VLCFA synthesis, residual ABCD1-mediated peroxisomal membrane translocation capacity, peroxisomal β--oxidation, and VLCFA elongation to longer species by ELOVL1[50]. In addition, dicarboxylic-VLCFA formation via ω-oxidation by CYP4F2 and CYP4F3B[51], followed by transport into peroxisomes by ABCD3[52], and VLCFA incorporation into complex lipids also influence VLCFA levels in ALD[53,54]. Changes in the enzymatic activity of key enzymes involved in these metabolic pathways have a direct impact on VLCFA homeostasis and levels. Indeed, pharmacological activation of stearoyl-CoA desaturase-1 (SCD1), the enzyme the mediates fatty acid saturation status, reduced *de novo* synthesis of saturated VLCFA[55]. Conversely, inhibition of SCD1 increased C26:0 levels[55]. VLCFA synthesis is primarily dependent on long-chain fatty acid elongation, which is mainly regulated by ELOVL1[50]. Pharmacological inhibition of ELOVL1 reduces C26:0 synthesis and levels[56]. Similarly, genetic variants in key enzymes of VLCFA homeostasis can influence VLCFA levels. For example, the p.V433M variant of CYP4F2 reduces enzyme levels, resulting in decreased conversion of VLCFA to VLCFA-dicarboxylic acids via ω--oxidation[57]. Saturated levels of VLCFA in complex lipids have previously been linked to a risk for leukodystrophy. For example, the accumulation of saturated VLCFA in the normal-appearing white matter of patients with cerebral ALD is higher than in individuals without cerebral involvement[58]. Additionally, lower levels of the monounsaturated fatty acids C22:1 and C24:1 were found in patients with cerebral ALD[58]. Importantly, high levels of saturated VLCFA induce oxidative stress, mitochondrial dysfunction, and an increased endoplasmic reticulum (ER) stress response[55,59-62]. In contrast, monounsaturated VLCFA do not cause these effects[55,59] and may even attenuate ER stress induced by high levels of saturated VLCFA[63]. Several studies have shown that VLCFA in the brain is predominantly located in phospholipids, which may be an important factor in the development of cerebral ALD[20,64]. This is supported by the demonstration that incorporation of VLCFA into artificial phospholipid vesicles resulted in disruption of membrane structure and function[27]. Interestingly, in our study, the lipid classes with the most significant fold increase in cerebral ALD were VLCFA-containing PC and LPC. Importantly, the most pronounced differences were found in saturated LPC species.

LPC and PC have been associated with cerebral ALD in several studies. Analysis of the lipid profile in postmortem brains from cerebral ALD cases revealed a significant increase in VLCFA-containing cholesterol esters in active demyelinating areas. Notably, in both the active demyelinating area and the intact white matter, VLCFA was prominently enriched in the PC fraction[20]. This indicates that the accumulation of VLCFA in PC precedes the onset of demyelination. Furthermore, VLCFA-enriched LPC are cytotoxic, as intracortical injection of LPC(24:0) induced widespread microglial activation and apoptosis in wild-type mice[28]. Importantly, this effect was absent with LPC(16:0) injection, underscoring that saturated VLCFA incorporation into lipids results in enrichment of cytotoxic lipid variants. Furthermore, a recent study reported decreased serum levels of LPC(20:3) and LPC(20:4) in CALD patients compared to asymptomatic ALD patients[65]. Our results confirm the importance of chain length and saturation status in the development and severity of ALD pathology.

In addition to their association with cerebral ALD, complex lipid VLCFA levels also affect non-neurological tissues. Here, we demonstrate a correlation between VLCFA levels in different lipid classes, including LPC, PC, and TG, and the onset of adrenal insufficiency. This difference became more pronounced when focusing on patients without adrenal insufficiency who were aged >55 years, a group with a lower risk of developing adrenal insufficiency[11]. Interestingly, previous research has shown a correlation between the presence of adrenal insufficiency in individuals with a Zellweger spectrum disorder and the plasma total C26:0 concentrations[66]. The exact mechanism by which VLCFA accumulation affects adrenal gland function remains not fully understood[6]. However, it has been suggested that VLCFA accumulation in CE may lead apoptosis and atrophy of the adrenal cortex, thereby disrupting cortisol production[67]. Furthermore, the physiological effect of VLCFA on cortisol release by human adreno-cortical cells was evident when VLCFA were added to the cell culture medium at concentrations similar to those found in ALD plasma. This resulted in increased membrane microviscosity and decreased ACTH-stimulated cortisol secretion, suggesting that excess VLCFA altered membrane structure and interfered with ACTH receptor availability[68]. Considering this, ALD patients with chronically elevated VLCFA levels may be at greater risk of developing adrenal insufficiency due to adrenal gland dysfunction.

In addition to cerebral ALD and adrenal insufficiency, virtually all adult male ALD patients eventually develop spinal cord disease. Our lipidomics results showed that patients with severe spinal cord disease symptoms (EDSS > 6) had higher levels of VLCFA-containing lipids compared to patients with mild spinal cord disease (EDSS ≤6 and >55 years of age). VLCFA has been implicated in the pathogenesis of ALD-related spinal cord disease. The abnormal concentration of VLCFA in myelin lipids of the spinal cord is associated with oxidative stress and impaired mitochondrial function[60], which likely contribute to spinal cord disease by disrupting ATP-dependent axonal transport. Moreover, lumbar dorsal root ganglia from ALD patients with spinal cord disease show neuronal atrophy

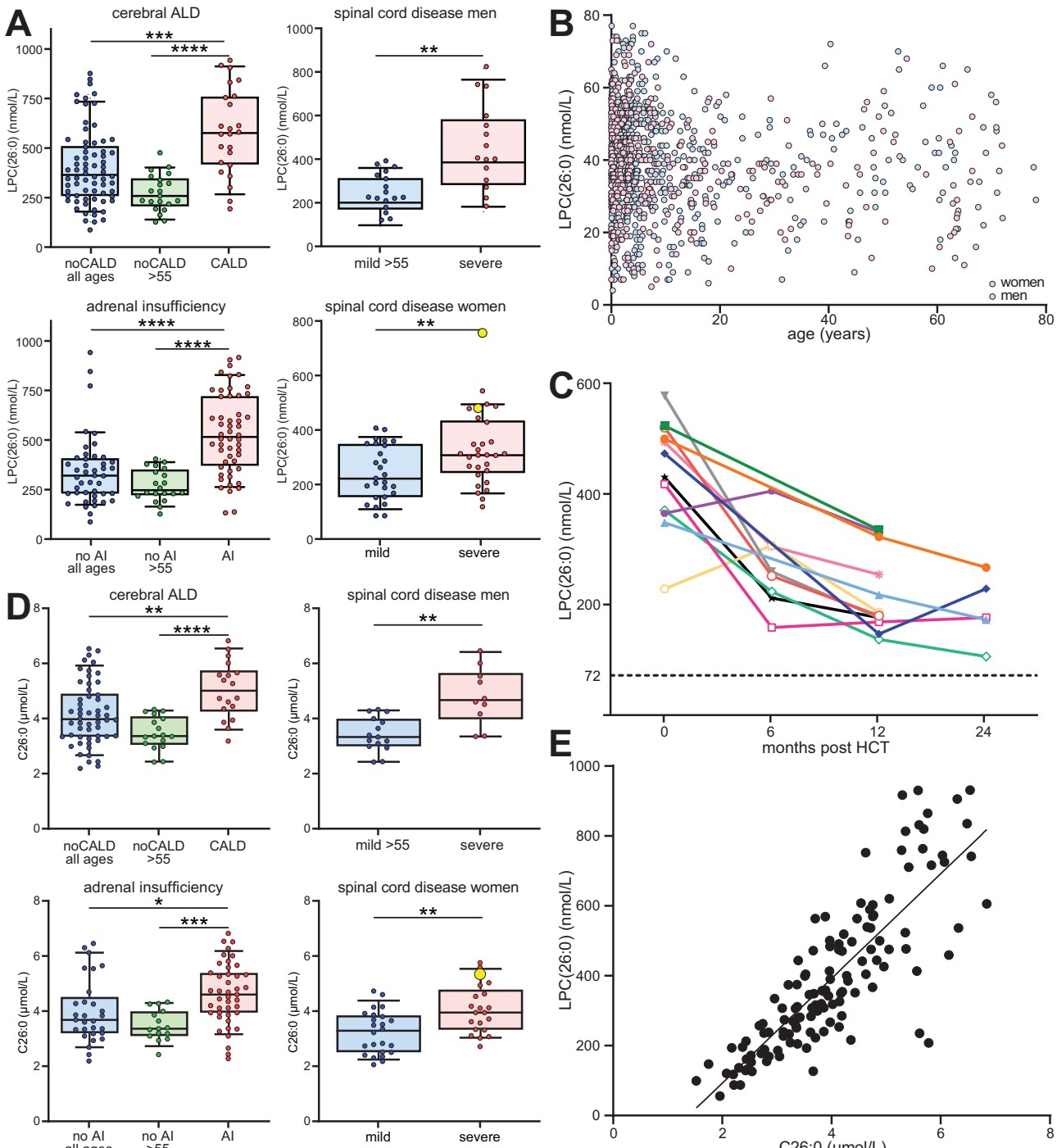

**Fig. 6 | Targeted analysis of LPC(26:0) and C26:0 in plasma. A** Box plots (whiskers are determined using Tukey) of plasma C26:0-lysophosphatidylcholine (LPC(26:0)) concentrations for different patient groups (male ALD patients $n = 112$, female ALD patients $n = 66$). Male patients with cerebral ALD (CALD) ($n = 25$), patients without CALD (noCALD) (all ages, $n = 71$), noCALD >55 years ($n = 21$), patients with adrenal insufficiency (AI) ($n = 53$) and patients without AI (NoAI) (all ages, $n = 43$), noAI >55 years ($n = 20$), patients with mild (EDSS≤6) spinal cord disease aged >55 years ($n = 17$) and patients with severe (EDSS > 6) spinal cord disease ($n = 15$). Female patients with mild (EDSS ≤ 6) spinal cord disease aged >55 years ($n = 27$) and patients with severe (EDSS>6) spinal cord disease ($n = 29$). Yellow dots represent

female patients ($n = 2$) with cerebral ALD. **B** Plasma LPC(26:0) concentration and age of 1090 controls. **C** Box plots (whiskers are determined using Tukey) of plasma LPC(26:0) concentration of 12 male CALD patients who underwent hematopoietic stem cell transplantation (HCT) at baseline and at different time points after the procedure. Each color represents a different patient. **D** Plasma C26:0 concentrations for different patient groups. The yellow dot represents the female patient with cerebral ALD. **E** Linear correlation between plasma LPC(26:0) and C26:0 concentrations (Spearman rank correlation coefficient of 0.7). Welch's $t$-test or Mann–Whitney $U$ test was used to determine significant differences between groups (*$P \leq 0.05$; **$P \leq 0.01$; ***$P \leq 0.001$; ****$P \leq 0.0001$).

and significant shifts in morphometrics, particularly in neuron size distribution[69]. Additionally, the detection of mitochondria containing lipid inclusions within neurons suggests impaired mitochondrial function. Furthermore, Gong et al. demonstrated that elevated phagocytosis markers,

including MFGE8 and TREM2, precede synapse loss in the spinal cord of ALD patients. Interestingly, the addition of LPC(26:0) to ABCD1-deficient microglia further increased MFGE8 expression, exacerbated phagocytosis and neuronal injury[70].

In contrast to male patients, female ALD patients typically develop spinal cord disease later in life often in their fifth decade or later. All male ALD patients have elevated plasma VLCFA levels, but 10–15% of female ALD patients have normal plasma C26:0 levels[19]. Previous research has shown that LPC(26:0) has superior diagnostic accuracy compared to total C26:0. Importantly, LPC(26:0) levels are elevated in the plasma of women with ALD who have normal plasma C26:0 levels[38,71]. Our lipidomics results support this, with no overlap in the relative abundance of LPC(26:0). In addition, our study identified several other elevated lipid markers in affected women with ALD that did not overlap with controls, such as CE(26:0) and PC(44:2). Furthermore, our results established a correlation between the severity of spinal cord disease in female patients and VLCFA-containing lipids, with the highest levels associated with severe disease. Notably, the two women with cerebral ALD had LPC(26:0) concentrations similar to those found in male patients with cerebral ALD.

Our study showed a robust correlation between plasma LPC(26:0) concentration and X-inactivation patterns. Notably, the ALD woman with cerebral ALD who was included in the X-inactivation analysis had the highest degree of skewing. These findings are consistent with previous research demonstrating a clear relationship between biochemical abnormalities in fibroblasts and X-inactivation patterns[13]. Specifically, when X-inactivation was favored in the allele harboring the pathogenic variant, fibroblasts exhibited more pronounced biochemical abnormalities. This strongly suggests a direct correlation between X-inactivation, disease severity, and neurological manifestations in women with ALD. Previous studies have yielded inconsistent results regarding the relationship between X-inactivation and symptomatic status. For example, Watkiss et al. found no association between the X-inactivation pattern in lymphocytes and symptoms; however, this study was limited by a small sample size ($n = 12$)[47]. The results reported by Maier et al. suggested an association between disease symptoms and X-inactivation pattern in leukocytes[46]. However, Salsano et al.

investigating X-inactivation patterns in peripheral blood mononuclear cells did not find a correlation between X-inactivation status and symptomatic status[72]. A limitation of their study was the age difference of >15 years between the asymptomatic and symptomatic groups, as age strongly predicts symptomatic status in women with ALD[13], making comparisons difficult.

Targeted LPC(26:0) analysis is critical for the diagnosis of ALD and newborn screening, providing accurate concentration values for robust comparisons between analytical runs and long-term monitoring of ALD patients. In our study, we complemented lipidomics with plasma LPC(26:0) quantification. Elevated LPC(26:0) concentrations correlated with cerebral ALD, adrenal insufficiency, and severe spinal cord disease in male ALD patients. By evaluating LPC(26:0) in 1064 control subjects, we established that neither age nor sex affected LPC(26:0) concentrations. We also evaluated LPC(26:0) changes in 12 male patients with cerebral ALD after HCT and found a significant and sustained reduction over time, demonstrating the efficacy of HCT in lowering plasma VLCFA-lipids. However, plasma LPC(26:0) levels did not fully normalize compared to healthy controls. In contrast to our results, van Geel et al. reported that HCT had an unclear effect on plasma VLCFA levels[73]. A limitation of their study was the small sample size and the lack of information on the timing of post-transplant samples, making direct comparisons difficult. Interestingly, the study by van Geel et al. shows that HCT does not prevent the occurrence of spinal cord disease, as 3 out of 5 patients developed signs of myelopathy in their twenties. Whether HCT (slightly) modifies disease progression remains unknown, as this would require decades of follow-up of a large number of patients. Our targeted LPC(26:0) findings are consistent with lipidomics, suggesting its potential as an accessible tool to assess the risk for cerebral ALD. Notably, in our lipidomic dataset, 1-acyl LPC(32:0) showed a more pronounced statistical difference compared to LPC(26:0) in different patient groups. Further research is needed to evaluate LPC(32:0) as a potential biomarker for ALD. Interestingly, the targeted analysis of LPC(26:0) by MSMS as described by Hubbard et al. could be modified to include 1-acyl LPC(32:0)[45].

Previous studies have concluded that plasma VLCFA analysis is not predictive of disease progression or severity in ALD[17–19]. Given the associations of plasma VLCFA-lipids and LPC(26:0) concentrations with disease severity in our study, we sought to further investigate the potential correlation between total plasma VLCFA concentrations and disease severity. Notably, total C26:0 fatty acid levels corroborated the results of lipidomics and targeted LPC(26:0) analysis. Elevated C26:0 levels were associated with the presence of cerebral ALD, adrenal insufficiency, and severe spinal cord disease in male ALD patients. The fact that this correlation has not been observed in previous studies is probably due to the confounding effect of disease progression over time described above. The availability of the large prospective cohort of well-characterized patients and associated biobank for the current project, as well as the classification by symptoms and age based on the current understanding of the natural history allowed the identification of the association between plasma VLCFA and disease severity. Moreover, in our study, we quantified total C26:0 levels using electrospray ionization mass spectrometry instead of the gas chromatography (GC) based method used in most studies[39,74]. It cannot be excluded that this difference in analytical

**Table 2 | The observed concentration ranges for the different patient groups**

| Group | LPC(26:0) nmol/L | | | C26:0 µmol/L | | |
|---|---|---|---|---|---|---|
| | Mean | Median | Range | Mean | Median | Range |
| CALD | 601 | 571 | 190−928 | 5 | 5 | 3.2−6.8 |
| noCALD | 395 | 356 | 85−863 | 4.2 | 4 | 2.2−6.5 |
| noCALD >55 | 269 | 254 | 125−468 | 3.5 | 3.4 | 2.4−4.3 |
| AI | 536 | 516 | 130−928 | 4.6 | 4.6 | 2.3−6.8 |
| NoAI | 341 | 316 | 85−928 | 4 | 3.7 | 2.2−6.5 |
| NoAI >55 | 267 | 242 | 125−398 | 3.5 | 3.4 | 2.4−4.3 |
| mild_males | 259 | 231 | 125−398 | 3.4 | 3.3 | 2.4−4.3 |
| severe_males | 472 | 411 | 189−832 | 4.8 | 4.7 | 3.4−6.5 |
| severe_females | 337 | 305 | 145−747 | 4.1 | 4 | 2.7−5.8 |
| mild_females | 235 | 213 | 85−402 | 3.2 | 3.3 | 2.1−4.7 |

**Table 3 | Significance and fold change for LPC(26:0), total C26:0 and LPC(32:0) for defined patient group comparisons**

| | LPC(26:0) | | C26:0 | | LPC(32:0) | |
|---|---|---|---|---|---|---|
| | $p$ | FC | $p$ | FC | $p$[a] | FC |
| CALD vs NoCALD (all ages) | 0.0001907 | 1.52 | 0.0067250 | 1.21 | 0.0000005 | 2.19 |
| CALD vs NoCALD >55 | 0.0000010 | 2.24 | 0.0002741 | 1.45 | 0.0000003 | 3.74 |
| AI vs NoAI (all ages) | 0.0000069 | 1.58 | 0.0106273 | 1.16 | 0.0000022 | 2.15 |
| AI vs NoAI >55 | 0.0000014 | 2.01 | 0.0008446 | 1.33 | 0.0000008 | 3.13 |
| severe vs mild (male) | 0.0005898 | 1.82 | 0.0013877 | 1.4 | 0.0007270 | 2.58 |
| severe vs mild (female) | 0.0076843 | 1.43 | 0.0064491 | 1.26 | na | na |

[a]Unadjusted $p$-value.

methodology may have contributed to the identification of correlations between VLCFA and disease severity. We observed a strong correlation between plasma total C26:0 and LPC(26:0) concentrations. However, the comparison of total C26:0 showed less substantial differences between patient groups compared to LPC(26:0). Although C26:0 analysis traditionally plays an important role in the diagnosis of ALD it is important to emphasize that LPC(26:0) is the preferred marker. C26:0 analysis is time-consuming, labor-intensive, and prone to false-positive results due to factors such as diet or blood sampling conditions[75–78]. In addition, LPC(26:0) has better diagnostic performance compared to VLCFA analysis using GCMS[38]. Fasting mitigates dietary effects on total C26:0 levels but poses challenges for patients. In contrast, LPC(26:0) levels remain relatively stable over time with no evidence of dietary influence, simplifying sampling and reducing false positives. In contrast, LPC(26:0) in blood exists primarily as a component of cell membranes in cells such as erythrocytes[79]. Given that erythrocytes have a lifespan of 2–4 months, the concentration of LPC(26:0) is expected to remain stable over this period simplifying sampling and reducing false positives[80,81].

The availability of a large prospective cohort of well-characterized patients and associated biobank samples allowed us to investigate the relationship between lipidome and disease severity in ALD. Our results revealed a strong association between higher levels of VLCFA-containing lipids and the presence of cerebral ALD, adrenal insufficiency, and severe spinal cord disease in male ALD patients. In female ALD patients, VLCFA-lipid levels correlate with X-inactivation patterns, and higher levels are associated with more severe disease manifestations. Our findings are supported by the concordance of LPC(26:0) and total VLCFA analysis with the lipidomics results. Overall, this study provides robust evidence of a strong association between plasma VLCFA-lipid levels and disease severity in both male and female ALD patients.

## Data availability

Lipidomics data are available as supplemental data files. All other data are available from the corresponding author on reasonable request. The source data for Figs. 1, 2, 3, and 4 is in Supplementary Data 1. The source data for Fig. 5 is in Supplementary Data 3. The source data for Fig. 6 is in Supplementary Data 5 and Supplementary Data 7. Supplementary Data 1: Lipidomics data males. Supplementary Data 2: Statistical results lipidomics data males. Supplementary Data 3: Lipidomics data females. Supplementary Data 4: Statistical results lipidomics data females. Supplementary Data 5: Targeted LPC(26:0) data males and females. Supplementary Data 6: Statistical results targeted LPC(26:0) data males and females. Supplementary Data 7: VLCFA data males and females. Supplementary Data 8: Statistical results VLCFA data males and females.

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

## Acknowledgements

The authors thank all ALD patients for their willingness to participate in this study. This research was funded by grants from the Netherlands Organization for Scientific Research, grant number 016.196.310 to M.E.; and the Netherlands Organization for Health Research and Development, grant number 543002004 to S.K.

## Author contributions

Y.J., M.E., and S.K. conceived the idea and planned the study. H.Y., C.B., I.H., M.V., R.T., T.L., W.K., M.E. performed clinical assessment. Y.J., I.D., and J.H. performed targeted assays and lipidomics experiments. G.S., M.E., and S.K. supervised the study. Y.J., E.W., A.J., F.V., M.E., and S.K. analyzed and interpreted the data. Y.J., M.E., and S.K. wrote the original draft. All authors contributed, revised, and approved the final version of the manuscript.

## Competing interests

The authors declare no competing interests.

## Additional information

¹Laboratory Genetic Metabolic Diseases, Department of Laboratory Medicine, Amsterdam UMC location University of Amsterdam, Amsterdam Neuroscience, Amsterdam Gastroenterology Endocrinology Metabolism, Amsterdam, The Netherlands. ²Department of Pediatric Neurology, Amsterdam UMC location University of Amsterdam, Amsterdam Leukodystrophy Center, Emma Children's Hospital, Amsterdam Neuroscience, Amsterdam, The Netherlands. ³Department of Neurology, Leukodystrophy Outpatient Clinic, Leipzig University Medical Center, Leipzig, Germany. ⁴Bioinformatics Laboratory, Department of Epidemiology and Data Science, Amsterdam Public Health Research Institute, Amsterdam UMC location University of Amsterdam, Amsterdam, The Netherlands. ⁵Core Facility Metabolomics, Amsterdam UMC location University of Amsterdam, Amsterdam, The Netherlands. ⁶Department of Pediatrics, Amsterdam UMC location University of Amsterdam, Emma Children's Hospital, Amsterdam, The Netherlands. ⁷Department of Pediatrics, Division of Bone Marrow Transplantation, University of Minnesota Children's Hospital, Minneapolis, MN, USA. ⁸These authors contributed equally: Marc Engelen, Stephan Kemp. ✉e-mail: s.kemp@amsterdamumc.nl

