## [Peer Review File · Communications Medicine]

Reviewers' comments:

Reviewer #1 (Remarks to the Author):

The study by Jaspers et al., “Lipidomic biomarkers in plasma correlate with disease severity in adrenoleukodystrophy” is focused on investigating a correlation between plasma lipidome and ALD disease severity. A longstanding problem in ALD has been the inability to predict the disease course especially in males with ALD. There is lack of genotype-phenotype correlation and ability to predict disease progression remains severely limited. The investigators of this study impressively employed a large multi-center cohort of patients to investigate association of plasma VLCFA-lipid levels with disease phenotype in ALD males and females.

I have several concerns with the study:

1. The authors state in line 121 “In this study, our main objective was to find lipids that correlate with disease course”. In my understanding the samples were collected at a single timepoint from patients whose disease status was already clinically established. Hence the study correlates lipid levels with disease expression and not with “disease course” that would require longitudinal samples pre- and post-appearance of clinical symptoms.
2. The authors state that “our results strongly suggest that VLCFA-containing lipids in plasma could serve as biomarkers for the risk of developing cerebral ALD”. Looking at Table2, while the median LPC and C26 values show a stratification the individual patient values have a broad range of variation. It is unclear how individual patient plasma LPC and C26 levels will be relevant as predictor of future cerebral disease in clinical settings for someone with lipid levels towards the lower spectrum of range but more severe disease or vice-versa.
3. The authors fail to acknowledge the documented role of inflammation-induced peroxisomal dysfunction resulting in increased VLCFA accumulation.
4. Line 424 should read “However, plasma LPC (C26:0) levels did NOT fully normalize compared to healthy controls.
5. While hormone replace therapy was mentioned, no information is provided if patients were on any other treatment regimen (Lorenzo’s oil), especially older patients who assumingly have been diagnosed for several years.
6. Given the variability in patient plasma lipid levels a posthoc test must be employed in addition to student t-test.

Reviewer #2 (Remarks to the Author):

In this manuscript, the authors conduct a lipidomic investigation to identify potential biomarkers for X-linked adrenoleukodystrophy (XLD). The study's cohort, comprising 167 adults, is notably large for a metabolic disorder of this nature. The research employs an unbiased approach in exploring the lipidome, a logical choice considering the emerging importance of LPC(C26:0) in newborn screening for adrenoleukodystrophy (ALD). Given the lipidome's high sensitivity to environmental influences and notable intraday fluctuations, the authors wisely apply component analysis and utilize well stratified cohort. This analysis reveals a correlation between the chain lengths of major lipid components and disease severity, categorizing severity using established scales and patient age.

The manuscript is articulate and effectively summarizes its key findings and their significance. However, it's important to note that the 1.4-1.9 logarithmic changes, particularly in the 1.4-1.6 range for the more critical indices as indicated in table 3, correspond to a 2.6-3.5-fold increase in total (2.6-3 in more critical indices). While statistically significant, these changes are not substantial enough to alter clinical decision-making dramatically in the immediate timeframe. Additionally, while LPC 26:0 and LPC 32:0 highlight differences across the six domains shown in the figures, other analytes like PC(48:4), PC(O-44:1), PC(46:2), SM(d44:1), LPC(28:1), TG(64:4) show inconsistent(?) elevations and do not consistently indicate elevated levels. This somewhat contrast the elegant demonstration of length-dependent increase. Then again, it also demonstrates that no other clear candidates can be promoted as sensitive and specific markers for CALD vs AMN.

Overall, the study offers a thorough examination of the LPC 26:0 biomarker among lipid alterations, strengthening the foundations on which it lies as a diagnostic marker adding to its potential in disease severity and monitoring. Along with 1-acyl LPC 32:0, it emerges as a prominent candidate in the search for untargeted markers. This research suggests that with a more carefully selected cohort, very-long-chain fatty acids (VLCFAs) might indeed correlate with disease severity, a vital consideration in the management of pre-symptomatically identified patients. Moreover, it makes a strong argument for including LPC 26:0 (and possibly 1-acyl LPC 32:0) in newborn screening and ongoing disease monitoring/managing.

I support the publication of this manuscript with addressing the following comments:

The "Methods" section of this manuscript is lacking critical details:

1. The subsection titled 'Lipidomics, LPC(26:0) and total VLCFA analysis in plasma' fails to briefly outline the procedure used, referencing only Vogel et al (ref #14), Valianpour et al (#37), and Sauteraud et al (#38), of which only #36 (Jaspers et al) seems relevant but lacks statistical discussion. Given the size of the cohort, the risk of Type 1 errors in such an unbiased search is high and should be meticulously addressed in the statistical approach.
2. It remains unclear how many analytes were examined in this study.

3. In the 'Statistics' section, the authors cite Mohamed et al (ref #41) regarding the lipidr R library. This reference describes the library as an automated parsing tool with intergroup comparison capabilities but does not detail the data processing method. Subsequently, reference #42, used for plotting, predates the creation of lipidr by 13 years, and discusses NBS, which raises questions about its relevance.

4. The authors state that they applied the BH FDR method for multiple testing within lipid classes but do not clarify if this was for all analyte comparisons or only specific subgroup comparisons. The latter approach could introduce statistical biases, although the figure captions and results suggest otherwise.

5. The rationale behind the size of the healthy control group (24 controls for 167 ALD subjects) is not specified. Was it determined by participation barriers, cost, or power calculations?

6. There is a discrepancy in the number of female participants in the study group, cited as 67 in the discussion and 68 in the abstract and methods.

Results section

1. Section “Cerebral ALD in males is associated with higher levels of VLCFA-containing lipids”

A. The volcano plot of CALD vs nonCALD show a scattering within a log fold change of 1 and -1. Despite p values within acceptable significance, these results are highly sensitive to biases. A more rigorous Bonferroni correction or permutation test is prudent to extract biologically meaningful data (or discussion thereof). Additionally, these small differences also translated into small variations in the ratio between LPC 26:0, PC(48:4) and in particular LPC 32:0 of CALD to non CALD.

B. B. The fold change differences in ALD and nonALD as carbon chains grow are marginal, with more noticeable differences only in patients >55 with the most carbon atoms.

C. Do the authors know, based on natural history and clinical experience, what % of patients is expected to transition from nonALD (all ages) to ALD based on the average age? Does it corroborate with the elevated LPC, PC, TG seen in nonALD vs nonALD >55?

D. Do we see similar lower polyunsaturated TG in nonALD >55 compared with ALD?

E. “These results could therefore suggest that ALD patients without cerebral involvement and high levels of VLCFA-containing lipids may be particularly susceptible to developing CALD” – this statement needs more statistical rigor. While this finding is of extreme interest, these three patients are not the ‘top 3’ – what happened to the other ‘high’ dots? Maybe tone down a little this statement?

2. Section “Adrenal insufficiency is associated with higher levels of VLCFA-containing lipids” – what can explain the difference between the observation that “virtually all male patients develop adrenal insufficiency” as published by the same group (here ref #55) and that only 50% of patients in the

cohort have no AI (with a mean age of 47 for those without AI), and that about 20% have no AI at age > 55?

3. Section “Disease severity in female ALD patients is correlated with VLCFA-lipid levels” –

A. PCA analysis shows no clear separation between controls and ALD patients.

B. The subdivision of mild vs severe disease does not show meaningful separation.

4. X-inactivation paragraph: VLCFA-containing lipids should be changed to LPC(26:0) as this is the only VLCFA that was measured.

5. Paragraph “Comparison of plasma VLCFA levels, LPC(26:0) and disease severity” – The discussion on total C26:0 and disease severity does not align with the referred to Figure 6D, which addresses post-HCT levels only.

Discussion section

1. The assertion that VLCFA-containing lipids strongly indicate a particular finding could be overstated and might benefit from toning down.

2. Pharmacological inhibition of ELOVL1 reduces C26:0 synthesis and levels (ref #55) – reference 55 does not deal with ELOVL1 inhibition. (Perhaps Ofman et al?)

3. Similarly, Fourcade et al (ref 56) does not deal with p.V433M variant of CYP4F. I think there is a frameshift of +2 between the in-text citation and the reference list.

4. Similarly, reference #63 and 64 do not address “Importantly, LPC are increasingly associated with cardiovascular diseases and neurodegenerative diseases, such as Alzheimer's disease”.

Furthermore, it is unclear to me that the discussion about putative role of LPC in Alzheimer's and CHD and the Lands cycle above it contributes to the manuscript, given its scope.

5. Discussion about post HCT levels – The contrast between the impact of HCT on VLCFA plasma levels and its failure to prevent AMN, as shown by van Geel, needs addressing, especially in light of the observed drop in VLCFA levels post-HCT in this study. According to the graph in figure 6C, the

levels pre and post HCT drop from 400-600 to 200-300, levels that correlates with the cohort's mild disease. However, van Geel's 3 patients that developed myelopathy post-transplant developed it in their 20s – a sign of (eventually) severe myelopathy, and this point should be addressed.

6. "Fasting mitigates dietary effects on total C26:0 levels but poses challenges for patients. In contrast, LPC(26:0) levels remain relatively stable over time, with no evidence of dietary influence, simplifying sampling and reducing false positives." – a reference is missing for this important remark.

Reviewer #3 (Remarks to the Author):

There has never been a measurement of lipid molecular species for each ALD symptom or female patients and carriers. If there are differences between them, this would be very interesting and an important finding; however, the manuscript presents the following significant concerns that currently hinder a detailed review of its content.

1) It should clearly state how many cases there are in the Netherlands, Germany, and the United States. In addition, the description of the number of patient samples used in the manuscript is inconsistent. The authors collected samples from 148 males and 68 females of ALD patients written in the method section; however, according to the summary in Table 1, the numbers of the samples used in Figs. 2, 3, and 5 are 99 males and 58 females. It is also unclear whether the count of the man, 33, in Fig. 4 is included in the above 99. These discrepancies should be clarified according to the classification of each group shown in 3 below for each country.

2) Were the samples were collected at the same facility and measured using the same equipment, or were they measured separately at three locations, and only the values were added together?

3) The categorization of patient groups into cases appears insufficient. In Fig. 2-4, cases are simply classified and analyzed based on the presence or absence of cerebral, adrenal, and spinal symptoms; however, since each patient may exhibit the presence or absence of each symptom, the categories could be divided into eight groups. Additionally, there might be a confounding effect among the symptoms of the cerebrum, adrenal glands, and spinal cord.

Alternatively, to analyze the effect of the cerebral symptoms on lipid molecular species, the following four categories could be defined: with both adrenal and spinal cord symptoms, with adrenal symptoms and without spinal symptoms, without adrenal symptoms with spinal cord symptoms and without both adrenal symptoms and spinal symptom. Within these four categories,

the authors can compare the groups with and without cerebral symptoms. Identifying common changes in molecular species accompanying cerebral symptoms in all four categories would highlight it as a characteristic of cerebral symptoms. The same approach can be applied to adrenal symptoms and spinal symptoms.

Moreover, it is essential to clarify whether spinal symptoms are limited to patients without cerebral symptoms. It is generally known that spinal type AMN may transition to cerebral-type, and if a patient with cerebral-type consistently exhibits spinal symptoms, the patient classification could be differentiated based on the severity of central nervous system as cerebral symptoms type, spinal cord symptoms type only, or no symptoms type. These can be combined with or without adrenal symptoms, resulting in six groups.

4) If the dataset does not include HCT except for specific cases, it should be explicitly mentioned.

5) The reason for creating the category for those over 55 is unclear. If age categorization is intended, it would be beneficial to subdivide the existing classification by age. Alternatively, analysis could be conducted exclusively for patients over 55 years of age in all categories. Even among those who have developed the disease, the values may be lower in people over 55 years of age. Naturally, controls should also be restricted to those over 55 years of age.

6) The absence of a description, along with references, detailing the lipidomics methodology significantly undermines the credibility of the paper. The reliability of this research hinges on the precise separation and quantification of individual lipid molecular species, raising concerns about the adequacy of the current methodology. It is essential to describe the methods of extraction, separation including column and solvents, detection and identification method including ion mode and the mode of mass spectrometer. From the figure, it appears that a wide variety of lipid classes were targeted; however what ranges of lipid classes and fatty acid chain lengths and number of unsaturation bonds were targeted? What are the internal standards? If measurements were conducted separately in the three countries, the authors should describe each measurement method and elucidate the measures taken to ensure consistency and guarantee identical values from the same sample.

7) A comparison between the non-symptomatic group and the control is required.

8) The vertical axis of the trend line is unspecified. Please provide the calculation formula. Additionally, how would the trend line appear regarding the unsaturated bond number?

9) In the analysis of each symptom, it is imperative to explicitly list the characteristic molecular species in the volcano plot as a table. Furthermore, by performing a volcano plot for each group and the control could reveal molecular species specific to each group.

10), Please provide a specification of the characteristics of molecular species for female patients, both in comparison to male patients as a whole and those with spinal symptoms.

11) It is essential to categorize patients undergoing HCT based on the degree of symptom improvement. Furthermore, variations in other molecular species obtained in above 9 should be described.

12) How was it determined that 32:0-LPC is the 1-acyl form? On the other hand, what about 26:0-LPC?

13) I would like to see measurements and considerations regarding the amount of molecules, not just increases and decreases. Is there a change in the composition, rather than a tiny amount of molecular fluctuation? Here, if you are only performing relative measurements instead of absolute quantification, you should confine the analysis to within-class analysis. The reason for this is that comparisons of signal intensities between different lipid classes are meaningless because their ionization efficiencies are completely different.

Re: Manuscript Number: COMMSMED-23-0845-T

Lipidomic biomarkers in plasma correlate with disease severity in adrenoleukodystrophy

Dear editor,

We would like to thank the reviewers for their detailed evaluation of the manuscript and their valuable comments. Below, we provide a point-by-point response and indicate whether changes have been made to sections of the manuscript.

Reviewers' comments:

Reviewer #1 (Remarks to the Author):

The study by Jaspers et al., "Lipidomic biomarkers in plasma correlate with disease severity in adrenoleukodystrophy" is focused on investigating a correlation between plasma lipidome and ALD disease severity. A longstanding problem in ALD has been the inability to predict the disease course especially in males with ALD. There is lack of genotype-phenotype correlation and ability to predict disease progression remains severely limited. The investigators of this study impressively employed a large multi-center cohort of patients to investigate association of plasma VLCFA-lipid levels with disease phenotype in ALD males and females.

I have several concerns with the study:

1.1 The authors state in line 121 "In this study, our main objective was to find lipids that correlate with disease course". In my understanding the samples were collected at a single timepoint from patients whose disease status was already clinically established. Hence the study correlates lipid levels with disease expression and not with "disease course" that would require longitudinal samples pre- and post-appearance of clinical symptoms.

Response: We thank the reviewer for the critical review of our work and suggested improvements. We acknowledge the reviewer's point regarding the terminology used in line 121. Although "disease course" could be inferred from the data, samples and associated clinical status reflect a single point in time for each included patient. Recognizing this, we now use "disease severity" throughout the manuscript.

1.2 The authors state that "our results strongly suggest that VLCFA-containing lipids in plasma could serve as biomarkers for the risk of developing cerebral ALD". Looking at Table2, while the median LPC and C26 values show a stratification the individual patient values have a broad range of variation. It is unclear how individual patient plasma LPC and C26 levels will be relevant as predictor of future cerebral disease in clinical settings for someone with lipid levels towards the lower spectrum of range but more severe disease or vice-versa.

Response: The data show that at the group level LPC(C26:0) is higher in those that develop cerebral ALD. The assumption is that this means that for an individual a high level of LPC(26:0) confers a higher risk of cerebral ALD. However, since there are other (genetic and environmental) factors that influence the risk of cerebral ALD, it is at this time not a marker that will guide clinical decision making. The current study shows for the first time that the degree of LPC(26:0) accumulation is important in disease expression and opens the way for the discovery of important pathophysiological mechanisms and a disease prediction model in the future, building on this important observation.

1.3 The authors fail to acknowledge the documented role of inflammation-induced peroxisomal dysfunction resulting in increased VLCFA accumulation.

Response: We apologize, but that it is not clear to us to which specific paper or study the reviewer is referring to regarding the documented role of inflammation-induced peroxisomal dysfunction and its association with VLCFA accumulation?

1.4. Line 424 should read “However, plasma LPC (C26:0) levels did NOT fully normalize compared to healthy controls.

Response: Thank you for highlighting this oversight. We have corrected the text to the following in the discussion section: “We also investigated LPC(26:0) changes 12 male cerebral ALD patients after HCT, revealing a significant and sustained reduction in over time, demonstrating HCT’s efficacy in lowering plasma VLCFA-lipids. However, plasma LPC(26:0) levels did not fully normalize compared to healthy controls.”

1. 5. While hormone replace therapy was mentioned, no information is provided if patients were on any other treatment regimen (Lorenzo’s oil), especially older patients who assumingly have been diagnosed for several years.

Response: Lorenzo’s oil (LO) treatment is a possible confounder and is not used in the Netherlands. We assumed that no patients were using LO. Upon receiving the reviewer comments, we contacted our collaborators from Germany and the U.S. to verify that no patients were on LO. Unfortunately, 9 patients from the German cohort were indeed using this. Since the effect on LO on plasma VLCFA (total C26:0) is a potential confounder, we excluded these patients. Following the exclusion of these 9 patients, we reanalyzed all the lipidomics data and updated all the figures accordingly. Fortunately, the exclusion had no effect on the results or conclusion of the study. We thank the reviewer for this point, since these patients should not have been included. The Section “Patient selection and clinical assessment” section now includes: “Patients did not receive Lorenzo’s oil.”

1.6 Given the variability in patient plasma lipid levels a posthoc test must be employed in addition to student t-test.

Response: We agree with the reviewer's point regarding the necessity of a post-hoc test, given the extensive range of lipids measured. To address this, we use the Benjamini-Hochberg (B&H) post hoc adjustment, which is the most used method in metabolomics and lipidomics field. This method offers a balanced approach to control for false discoveries while accommodating the complexity and correlation within lipidomics data. We have expanded the bioinformatics and statistics section of our manuscript to clarify data processing, statistical analysis and post-hoc B&H adjustment: “Statistical analyses were conducted utilizing R version 4.0.2 and Prism version 9.5.1. The normality of the data distribution was assessed using a Shapiro-Wilk test. Depending on the distribution, either a Welch's t-test or a Mann-Whitney U test was employed to compare groups. To correct for multiple testing, a post hoc Benjamini–Hochberg adjustment was applied to the p-values on a per comparison basis. Significance was established with a threshold of $p < 0.05$. Correlation between continuous variables was investigated using the two-tails Spearman correlation test.”

Reviewer #2 (Remarks to the Author):

In this manuscript, the authors conduct a lipidomic investigation to identify potential biomarkers for X-linked adrenoleukodystrophy (XLD). The study's cohort, comprising 167 adults, is notably large for a metabolic disorder of this nature. The research employs an unbiased approach in exploring the lipidome, a logical choice considering the emerging importance of LPC(C26:0) in newborn screening for adrenoleukodystrophy (ALD). Given the lipidome's high sensitivity to environmental influences and notable intraday fluctuations, the authors wisely apply component analysis and utilize well stratified cohort. This analysis reveals a correlation between the chain lengths of major lipid components and disease severity, categorizing severity using established scales and patient age.

The manuscript is articulate and effectively summarizes its key findings and their significance. However, it's important to note that the 1.4-1.9 logarithmic changes, particularly in the 1.4-1.6 range for the more critical indices as indicated in table 3, correspond to a 2.6-3.5-fold increase in total (2.6-3 in more critical indices). While statistically significant, these changes are not substantial enough to alter clinical decision-making dramatically in the immediate timeframe. Additionally, while LPC 26:0 and LPC 32:0 highlight differences across the six domains shown in the figures, other analytes like PC(48:4), PC(O-44:1), PC(46:2), SM(d44:1), LPC(28:1), TG(64:4) show inconsistent(?) elevations and do not consistently indicate elevated levels. This somewhat contrast the elegant demonstration of length-dependent increase. Then again, it also demonstrates that no other clear candidates can be promoted as sensitive and specific markers for CALD vs AMN.

Overall, the study offers a thorough examination of the LPC 26:0 biomarker among lipid alterations, strengthening the foundations on which it lies as a diagnostic marker adding to its potential in disease severity and monitoring. Along with 1-acyl LPC 32:0, it emerges as a prominent candidate in the search for untargeted markers. This research suggests that with a more carefully selected cohort, very-long-chain fatty acids (VLCFAs) might indeed correlate with disease severity, a vital consideration in the management of pre-symptomatically identified patients. Moreover, it makes a strong argument for including LPC 26:0 (and possibly 1-acyl LPC 32:0) in newborn screening and ongoing disease monitoring/managing.

I support the publication of this manuscript with addressing the following comments:

The "Methods" section of this manuscript is lacking critical details:

2.1

The subsection titled 'Lipidomics, LPC(26:0) and total VLCFA analysis in plasma' fails to briefly outline the procedure used, referencing only Vogel et al (ref #14), Valianpour et al (#37), and Sauteraud et al (#38), of which only #36 (Jaspers et al) seems relevant but lacks statistical discussion. Given the size of the cohort, the risk of Type 1 errors in such an unbiased search is high and should be meticulously addressed in the statistical approach.

Response: We thank the reviewer for the support for publication and the critical review of our work. We agree with the reviewer regarding the necessity of taking into account type 1 errors. To address this, we have expanded the description of the statistics and the Benjamini-Hochberg (B&H) post hoc adjustment to the p-values to control the false discovery rate (see also response 1.6). We also expanded the methods section and now includes more detailed description of the lipidomics analysis, and the bioinformatics and statistics:

Lipidomics analysis

Lipidomics analysis was performed as previously described^{14,37}. In a 2 ml tube 20 µl of plasma was added and mixed with a mix of internal standards for different lipid classes, including 0.1 nmol of cardiolipin CL(14:0/14:0/14:0/14:0), 2.0 nmol of phosphatidylcholine PC(14:0/14:0), 0.1 nmol of

phosphatidylglycerol PG(14:0/14:0), 5.0 nmol of phosphatidylserine PS(14:0/14:0), 0.5 nmol of phosphatidylethanolamine PE(14:0/14:0), 0.5 nmol of phosphatidic acid PA(14:0/14:0), 2.125 nmol of sphingomyelin SM(d18:1/12:0), 0.02 nmol of lysophosphatidylglycerol LPG(14:0), 0.1 nmol of lysophosphatidylethanolamine LPE(14:0), 0.5 nmol of lysophosphatidylcholine LPC(14:0), 0.1 nmol of lysophosphatidic acid LPA(14:0), 0.5 nmol of phosphatidylinositol PI(8:0/8:0), 0.5 nmol diglycerides DG(14:0/14:0), 0.5 nmol triglycerides TG(14:0/14:0/14:0), 2.5 nmol cholesterol ester D₇-CE(16:0), 0.125 nmol of sphingosine and ceramide mix (Avanti Polar Lipids) dissolved in 1:1 (v/v) methanol:chloroform. Next, 1.5 ml 1:1 (v/v) methanol:chloroform was added to each sample. The mixture was sonicated in a water bath (5 min) and centrifuged (4 °C, (16,000×g, 10 min). The supernatant was transferred to a 1.5 ml glass auto sampler vial and evaporated under a stream of nitrogen at 45°C. The dried lipids were reconstituted in 100 µl of 1:1 (v/v) chloroform:methanol. Chromatographic separation of lipids was done using a Thermo Fisher Scientific Ultimate 3000 binary UPLC using a normal phase and a reverse phase column in separate runs. Normal-phase separation was done using a Phenomenex® LUNA silica, 250×2 mm, 5 µm 100 Å column. Column temperature was held constant at 25°C. The composition of the mobile phase A consisted of 85:15 (v/v) methanol:water containing 0.0125% formic acid and 3.35 mmol/l ammonia and the composition of mobile phase B consisted of 97:3 (v/v) chloroform:methanol containing 0.0125% formic acid. The LC gradient started at of 10% A for 0-1 min, 20% A at 4 min, 85% A at 12 min, 100% A at 12.1 min, 100% A for 12.1-14 min, 10% A at 14.1 min, 10% A for 14.1-15 min using a flow rate of 0.3 ml/min. Reversed-phase separation was done using a Waters HSS T3 column (150×2.1 mm, 1.8 µm particle size). The composition of the mobile phase A consisted of 4:6 (v/v) methanol:water and B 1:9 (v/v) methanol:isopropanol, both containing 0.1% formic acid and 10 mmol/l ammonia. The gradient started at 100% A going to 80% A at 1 min and 0% A at 16 min, 0% A for 16-20 min, 100% A at 20.1 min and 100% A for 20.1-21 min. The column temperature was held constant at 60°C and a flow rate of 0.4 ml/min was used. After LC separation, lipids were detected using a Q Exactive Plus Orbitrap mass spectrometer (Thermo Scientific) using negative and positive ionization. The spray voltage was 2,500 V and nitrogen was used as the nebulizing gas. A resolution of 280,000 was used in a mass range of m/z 150 to m/z 2,000.

Quantification of LPC(26:0)

Quantification of LPC(26:0) was performed as described earlier³⁸. Briefly, 10 µL plasma was extracted with 10 µL of an internal standard solution containing 1 µmol/L D4-C26:0-lysoPC in 0.5 mL of acetonitrile by ultrasonication for 5 min in a sonicator bath (Branson 3510) at room temperature. After centrifugation (5 min, 14000 RPM) the resulting methanol (DBS) or acetonitrile (plasma) layer was transferred to a new glass tube and evaporated under a constant stream of nitrogen at 40 °C. The samples were then reconstituted in 50 µL methanol, transferred to a sample vial, and capped. Samples were injected using an ACQUITY UPLC system (Waters, Milford, MA, USA) on a 50 x 2.1 mm, 2.6 µm particle diameter Kinetex C8 column (Phenomenex, Torrance, CA, USA). The column was held at a constant temperature of 50 °C. The composition of mobile phase A was 0.1% formic acid in water and mobile phase B was 0.1% formic acid in methanol. The gradient used was as follows: T = 0 min: 36% A, 64% B, flow 0.4 mL/min towards T = 6 min: 0% A, 100% B, flow 0.4 mL/min; T = 6–11 min: 0% A, 100% B, flow 0.4 mL/min, and T = 11–11.1 back to 36% A, 64% B, flow 0.4 mL/min. Detection was done using a Quattro Premier XE (Waters, Milford, MA, USA) using electrospray ionization in positive mode. The source temperature was 130 °C, and capillary voltage was 3.5 kV. Multiple reaction monitoring (MRM) was done on masses 636.50>104.10 and 640.50>104.10 with a dwell time of 0.03 seconds. Argon was used as a collision gas.

Bioinformatics and statistics

The raw LC/MS data were converted to mzXML format using MSConvert⁴². Lipidomics data processing and analysis was done as described earlier³⁷. Briefly, lipidomics data were analyzed using an in-house developed lipidomics pipeline based on the R programming language (<http://www.r-project.org>) and

MATLAB. Preprocessing was done using the R package XCMS with minor changes to some functions to better suit the Q Exactive™ data; notably, the definition of noise level in centWave was adjusted and the stepsize in fillPeaks⁴³. Lipid identification was based on a combination of accurate mass, (relative) retention times, analysis of samples with known metabolic defects, and the injection of relevant standards. Lipid classes were defined in our lipidomics pipeline in terms of their generic chemical formula, where R represents the radyl group. Upon import of the lipid database in the annotation pipeline, the generic chemical formula of each lipid class was expanded by replacing the R-element with a range of possible radyl group lengths and amount of unsaturations/double bonds. The resulting expanded list of chemical formulas was then used to calculate the neutral monoisotopic mass of each species. The monoisotopic mass was subsequently transformed to a set of m/z values for each adduct/charge combination with which that specific species could be reliably measured/annotated. The reported lipid abundances are semi-quantitative and calculated by dividing the response of the analyte (area of the peak) by that of the corresponding internal standard multiplied by the concentration of that internal standard (arbitrary unit, A.U). Statistical analyses were conducted utilizing R version 4.0.2 and Prism version 9.5.1. The normality of the data distribution was assessed using a Shapiro-Wilk test. Depending on the distribution, either a Welch's t-test or a Mann-Whitney U test was employed to compare groups. To correct for multiple testing, a post hoc Benjamini–Hochberg adjustment was applied to the p-values on a per comparison basis. Significance was established with a threshold of p<0.05. Correlation between continuous variables was investigated using the two-tails Spearman correlation test. Lipid saturation and chain length plots were generated using the R package lipidr version 2.15.1⁴⁴. In lipidr, chain length plots are created by plotting a regression line (LOESS curve) of the (log₂) fold changes and the total chain lengths of lipids within a specific lipid class.”

2.2.

It remains unclear how many analytes were examined in this study.

Response: Within our lipidomics study, we identified 1556 lipids. This information is provided in the section “ALD results in VLCFA incorporation in complex lipids”:

“Lipidomics analysis using plasma samples from 92 male ALD patients and 12 healthy male controls identified 1556 lipids and revealed that the lipidome of ALD patients was significantly different from that of controls, as indicated by the results of Principal Component Analysis (PCA) and Partial Least Squares Discriminant Analysis (PLS-DA) (Figure 1A).”

2.3.

In the 'Statistics' section, the authors cite Mohamed et al (ref #41) regarding the lipidr R library. This reference describes the library as an automated parsing tool with intergroup comparison capabilities but does not detail the data processing method. Subsequently, reference #42, used for plotting, predates the creation of lipidr by 13 years, and discusses NBS, which raises questions about its relevance.

Response: We corrected the references and have expanded the bioinformatics and statistics section of our manuscript to include the data processing method:

“The raw LC/MS data were converted to mzXML format using MSConvert⁴². Lipidomics data processing and analysis was done as described earlier³⁷. Briefly, lipidomics data were analyzed using an in-house developed lipidomics pipeline based on the R programming language (<http://www.r-project.org>) and MATLAB. Preprocessing was done using the R package XCMS with minor changes to some functions to better suit the Q Exactive™ data; notably, the definition of noise level in centWave was adjusted and the stepsize in fillPeaks⁴³. Lipid identification was based on a combination of accurate mass, (relative) retention times, analysis of samples with known metabolic defects, and the injection of relevant standards. Lipid classes were defined in our lipidomics pipeline in terms of their generic chemical formula, where R represents the radyl group. Upon import of the lipid database in the annotation pipeline, the generic chemical formula of each lipid class was expanded by replacing the R-element with a range of possible radyl group lengths and amount of unsaturations/double bonds. The resulting

expanded list of chemical formulas was then used to calculate the neutral monoisotopic mass of each species. The monoisotopic mass was subsequently transformed to a set of m/z values for each adduct/charge combination with which that specific species could be reliably measured/annotated. The reported lipid abundances are semi-quantitative and calculated by dividing the response of the analyte (area of the peak) by that of the corresponding internal standard multiplied by the concentration of that internal standard (arbitrary unit, A.U)."

2.4.

The authors state that they applied the BH FDR method for multiple testing within lipid classes but do not clarify if this was for all analyte comparisons or only specific subgroup comparisons. The latter approach could introduce statistical biases, although the figure captions and results suggest otherwise.

Response: In response to the reviewer's query regarding the application of the BH FDR method for multiple testing adjustments, we would like to clarify that the FDR correction was applied to each comparison separately. We have clarified the application of the FDR method in the bioinformatics and statistics section:

"To correct for multiple testing, a post hoc Benjamini–Hochberg adjustment was applied to the p-values on a per comparison basis"

To address potential concerns about statistical biases, we would like to emphasize that we do not base our conclusions on single or random lipids. Our lipidomics findings demonstrate that the elevations we observed in lipid levels are specific to lipids containing very long-chain fatty acids (VLCFA). Lipids of "normal length" exhibited no significant differences or showed reduced levels. This strongly indicates that our findings are not the result of statistical artifacts but align with the biological context of ALD. Moreover, It's important to emphasize that we further substantiated our lipidomics findings, using targeted analyses of LPC(26:0) using a validated diagnostic assay, alongside total VLCFA analysis. The outcomes of these targeted analyses were in close agreement with our initial lipidomics results.

2.5.

The rationale behind the size of the healthy control group (24 controls for 167 ALD subjects) is not specified. Was it determined by participation barriers, cost, or power calculations?

Response: The size of the healthy control group in our study was primarily influenced by the availability of control sample and the technical limitations with regards to how many samples can be included in a single lipidomics run which in addition to the control and patient samples also includes many quality-control samples. Additionally, our previous experience with LPC(26:0) analysis (Jaspers et al. 2020, doi: 10.3389/fcell.2020.00690) as well as the experience from our diagnostic laboratory with the LPC(26:0) analysis with >1000 analyses of controls (shown in Figure 6B) informed our decision, as our primary interest was in identifying large differences between ALD patients and controls rather than nuanced variations within these groups. Furthermore, based on our observations and experience, the variability within the control group tends to be relatively low.

2.6.

There is a discrepancy in the number of female participants in the study group, cited as 67 in the discussion and 68 in the abstract and methods.

Response: The reviewer is correct that the number of samples included in the lipidomics and LPC(26:0) are different. The lipidomics was performed using plasma samples from 65 female ALD patients. Of note, the current number is 65 instead of 67, because 2 female patients had to be excluded following the question regarding the use of Lorenzo's oil raised by reviewer 1 (please see 1.5 for details). To confirm the lipidomics findings, we performed targeted LPC(26:0) analysis for which we were able to include a plasma sample from an additional female with CALD sample bringing the total to 65+1 included females in total for the LPC(26:0) analysis. Because we realized this difference in number of samples included should be explained, we had clarified this in the Results section "Plasma

LPC(26:0) concentration determined by targeted assay correlates with disease severity in ALD patients”, with the statement: “Additionally, we obtained and included an extra plasma from a second female ALD patient with cerebral ALD.”

Results section

2.7.

Section “Cerebral ALD in males is associated with higher levels of VLCFA-containing lipids”
A. The volcano plot of CALD vs nonCALD show a scattering within a log fold change of 1 and -1. Despite p values within acceptable significance, these results are highly sensitive to biases. A more rigorous Bonferroni correction or permutation test is prudent to extract biologically meaningful data (or discussion thereof). Additionally, these small differences also translated into small variations in the ratio between LPC 26:0, PC(48:4) and in particular LPC 32:0 of CALD to non CALD.

Response: Within this comparison, it's crucial to recognize that some younger nonCALD patients are still at risk to develop CALD in the future. In addition to LPC(26:0) other factors (modifier genes, environmental factors) participate in the final clinical outcome of an individual patient. Furthermore, a log₂ fold change of -1 and 1 translates to fold changes of 0.5 and 2, respectively. This implies that for lipids with a fold change of 2, the average level in the CALD group is twice as high as that in the NoCALD group. Conversely, a fold change of 0.5 indicates that the average level in the CALD group is half that of the NoCALD group. We consider these differences to be relevant within the context of ALD.

In response to the reviewer's question about our choice of statistical correction, we would like to emphasize that the Bonferroni correction assumes that tests of features are independent and is often considered too strict in metabolomics and lipidomics studies. Especially in lipidomics, where datasets are high-dimensional and tests are not independent due to correlated lipids. This method significantly increases the risk of Type II errors, potentially missing biologically meaningful associations. Given the exploratory nature of lipidomics, the stringency of Bonferroni would hinder the discovery of insightful findings. Therefore, we chose the False Discovery Rate (FDR) method, the Benjamini-Hochberg procedure, which is the most widely used method in metabolomics and lipidomics. This method provides a balanced approach to control for false discoveries while taking into account the complexity and correlation within lipidomics data.

We would also like to point out that our lipidomics results are not random, but are consistent with the biological context of ALD. Specifically, lipids that contain VLCFA show elevated levels in CALD patients, whereas lipids lacking VLCFA either show no significant difference between CALD and NoCALD or are found at reduced levels in CALD. To support our lipidomics observations, we performed targeted analyses of LPC(26:0) using a validated and established diagnostic assay. The results of this targeted analysis were highly consistent with our lipidomics findings. Additionally, we performed targeted analyses of total VLCFA levels, which further confirmed the patterns identified in our lipidomics study and the LPC(26:0) targeted analysis. Given the consistency of these findings and the pathophysiological context of ALD, we are confident that our lipidomics results are not due to statistical bias.

2.8.

The fold change differences in ALD and nonALD as carbon chains grow are marginal, with more noticeable differences only in patients >55 with the most carbon atoms.

Response: Some patients in the noCALD (all ages) group are at risk to still develop cerebral ALD. Therefore, overlap is to be expected. Based on natural history, the risk to develop CALD reduces with age. Patients over 55 years of age have relatively low probability to convert to CALD. Therefore, our data show a bigger difference when comparing the CALD group to the noCALD >55 group.

2.9.

Do the authors know, based on natural history and clinical experience, what % of patients is expected to transition from nonALD (all ages) to ALD based on the average age? Does it corroborate with the elevated LPC, PC, TG seen in nonALD vs nonALD>55?

Response: Regarding the subgroup analysis of patients older than 55 years old, our rationale is based on existing natural history data, which indicates a relatively lower probability to develop cerebral ALD and adrenal insufficiency in this age group (DOI: 10.1210/jc.2018-01307). The probability of developing CALD decreases with age, but cases of men CALD at an advanced (>55 years) age have been described. We have chosen 55 years of age as a somewhat pragmatic cut-off for this group.

2.10

Do we see similar lower polyunsaturated TG in nonALD>55 compared with ALD?

Response: We observed a decrease in polyunsaturated TG levels in patients CALD compared to noCALD (all ages). As can be seen in Figure 2C, when comparing noCALD>55 (green line) to CALD (red line) this difference becomes more pronounced.

2.11

“These results could therefore suggest that ALD patients without cerebral involvement and high levels of VLCFA-containing lipids may be particularly susceptible to developing CALD” – this statement needs more statistical rigor. While this finding is of extreme interest, these three patients are not the ‘top 3’ – what happened to the other ‘high’ dots? Maybe tone down a little this statement?

Response: We agree with the reviewer that these three patients are of great interest. Follow-up studies on other “high” patients without CALD are ongoing, but it may take years for them to develop cerebral ALD. As suggested by the reviewer we have expanded the following sentence in the section “Cerebral ALD in males is associated with higher levels of VLCFA-containing lipids”:

“These results could therefore suggest that ALD patients without cerebral involvement and high levels of VLCFA-containing lipids may be particularly susceptible to developing CALD. Further studies in large cohorts of patients with samples before and after the onset of cerebral ALD are essential to better understand how VLCFA-containing lipid levels influence the risk of developing CALD.”

2.12

Section “Adrenal insufficiency is associated with higher levels of VLCFA-containing lipids” – what can explain the difference between the observation that “virtually all male patients develop adrenal insufficiency” as published by the same group (here ref #55) and that only 50% of patients in the cohort have no AI (with a mean age of 47 for those without AI), and that about 20% have no AI at age > 55?

Response: The statement “virtually all male patients develop adrenal insufficiency” is from an earlier publication from 2017. Of note, the complete statement in that publication is “Virtually all male patients develop adrenal insufficiency and myelopathy (adrenomyeloneuropathy), ...”. We acknowledge that in the 2017 paper it would have been more precise to write “and/or” instead of “and”. A subsequent retrospective natural history study of adrenal insufficiency that was published in 2019 showed that the lifetime prevalence of adrenal insufficiency in male patients with ALD is approximately 80% (DOI: 10.1210/jc.2018-01307) (Reference #11).

2.13

Section “Disease severity in female ALD patients is correlated with VLCFA-lipid levels” – A. PCA analysis shows no clear separation between controls and ALD patients.

Response: While it's accurate that the PCA plot does not demonstrate full separation of control subjects and female ALD patients, our analysis does reveal grouping that is indicative of differential lipid profiles. It's important to acknowledge that female ALD patients exhibit a less pronounced

difference from controls compared to male ALD patients. This variance can be attributed to the compensatory effect of the normal X chromosome in females, which mitigates the biochemical impact of ALD and results in a less severe phenotype. An example of this can also be seen in Figure 3C; some females have VLCFA-lipid levels in the control range.

2.14

The subdivision of mild vs severe disease does not show meaningful separation.

Response: The reviewer is correct that in our lipidomics analysis (Figure 5), the distinction between mild and severe disease indeed presents a modest, yet statistically not significant difference. However, when LPC(26:0) concentrations are determined using the targeted LPC(26:0) analysis (Fig 6A), the separation between mild and severe disease is statistically significant. We updated the text to more accurately reflect these nuances:

“The lipidomics analysis revealed several moderately elevated lipids that were associated with the severe group compared to the mild group, which included VLCFA-containing species from the classes LPC, PC, and TG (Figure 5C).”

2.15

X-inactivation paragraph: VLCFA-containing lipids should be changed to LPC(26:0) as this is the only VLCFA that was measured.

Response: We agree with the reviewer and corrected the text in the section “X-inactivation pattern correlates with LPC(26:0) levels in female ALD patients”:

“Taken together, these findings demonstrate a strong connection between the X-inactivation and LPC(26:0) levels.”

2.16

Paragraph “Comparison of plasma VLCFA levels, LPC(26:0) and disease severity” – The discussion on total C26:0 and disease severity does not align with the referred to Figure 6D, which addresses post-HCT levels only.

Response: We apologize, but we are not sure what the reviewer is referring to. The figure that addresses post-HCT levels is Figure 6C and the figure addressing the correlation between plasma VLCFA and disease severity is Figure 6D and the comparison between VLCFA and LPC(26:0) is Figure 6E.

Discussion section

2.17

The assertion that VLCFA-containing lipids strongly indicate a particular finding could be overstated and might benefit from toning down.

Response: We updated the text to a more nuanced phrased assertion:

“Taken together, our results suggest that VLCFA-containing lipids in plasma may be indicative of the risk of developing cerebral ALD. However, due to the diversity of VLCFA lipid levels observed, accurate predictions at the individual level remain challenging at present.”

2.18

Pharmacological inhibition of ELOVL1 reduces C26:0 synthesis and levels (ref #55) – reference 55 does not deal with ELOVL1 inhibition. (Perhaps Ofman et al?)

Response: Thank you for highlighting this oversight. We have corrected the references.

2.19

Similarly, Fourcade et al (ref 56) does not deal with p.V433M variant of CYP4F. I think there is a frameshift of +2 between the in-text citation and the reference list.

Response: Thank you for highlighting this oversight. We have corrected the references.

2.20

Similarly, reference #63 and 64 do not addresses “Importantly, LPC are increasingly associated with cardiovascular diseases and neurodegenerative diseases, such as Alzheimer’s disease”. Furthermore, it is unclear to me that the discussion about putative role of LPC in Alzheimer’s and CHD and the Lands cycle above it contributes to the manuscript, given its scope.

Response: We agree with the suggestion of the reviewer and removed the mentioned text.

2.21

Discussion about post HCT levels – The contrast between the impact of HCT on VLCFA plasma levels and its failure to prevent AMN, as shown by van Geel, needs addressing, especially in light of the observed drop in VLCFA levels post-HCT in this study. According to the graph in figure 6C, the levels pre and post HCT drop from 400-600 to 200-300, levels that correlates with the cohort’s mild disease. However, van Geel’s 3 patients that developed myelopathy post-transplant developed it in their 20s – a sign of (eventually) severe myelopathy, and this point should be addressed.

Response: Indeed, the study by van Geel et al. shows that HCT does not prevent myelopathy. In their study, 3 out of 5 patients developed signs of myelopathy in their twenties. Whether HCT (slightly) modifies the progression or occurrence of myelopathy cannot be excluded, but it is also very difficult to prove, as the required study would be very challenging and time consuming. We have modified the Discussion section and added the following:

“In contrast to our findings, van Geel et al. reported that HCT had an unclear effect on plasma VLCFA levels (72). A limitation of their study was the small sample size and the lack of the post-transplant sample timing information, making direct comparisons challenging. Interestingly, the study by van Geel et al. shows that HCT does not prevent the occurrence of spinal cord disease, as 3 out of 5 patients developed signs of myelopathy in their twenties. Whether HCT (slightly) modifies disease progression remains unknown, as this would require decades of follow-up of a large number of patients.”

2.22

“Fasting mitigates dietary effects on total C26:0 levels but poses challenges for patients. In contrast, LPC(26:0) levels remain relatively stable over time, with no evidence of dietary influence, simplifying sampling and reducing false positives.” – a reference is missing for this important remark.

Response: We have updated the text to include the following:

“In contrast, LPC(26:0) in blood primarily exists as a component of cell membranes in cells such as erythrocytes⁷⁸. Considering that erythrocytes have a lifespan ranging from 2 to 4 months, the concentration of LPC(26:0) is expected to remain stable during this period which simplifies sampling and reduces false positives^{79,80}.”

Reviewer #3 (Remarks to the Author):

There has never been a measurement of lipid molecular species for each ALD symptom or female patients and carriers. If there are differences between them, this would be very interesting and an important finding; however, the manuscript presents the following significant concerns that currently hinder a detailed review of its content.

3.1

It should clearly state how many cases there are in the Netherlands, Germany, and the United States. In addition, the description of the number of patient samples used in the manuscript is inconsistent. The authors collected samples from 148 males and 68 females of ALD patients written in the method section; however, according to the summary in Table 1, the numbers of the samples used in Figs. 2, 3, and 5 are 99 males and 58 females. It is also unclear whether the count of the man, 33, in Fig. 4 is included in the above 99. These discrepancies should be clarified according to the classification of each group shown in 3 below for each country.

Response: We extend our gratitude to the reviewer for the feedback on our work. In response to the reviewer's comment, we have made updates to the "Patient Selection and Clinical Assessment" section, specifically incorporating of participants from each country:

"Plasma samples were collected from 24 healthy controls (12 male and 12 female) and 178 ALD patients (112 male and 66 female) from the biobank linked to the "Dutch ALD cohort" (130 patients), the German center of excellence for ALD in Leipzig (32 patients) and the university of Minnesota Division of Pediatric Blood and Marrow Transplant and Cellular Therapies (16 patients)³²"

In this study, lipidomics analysis was conducted 92 male and 65 female ALD patients. (7 male and two female patients had to be excluded following the question regarding the use of Lorenzo's oil raised by reviewer 1 (please see 1.5 for details). For the lipidomics comparison of female ALD patients to controls, all 65 female patients were included. In addition, as detailed in the section "Patient stratification lipidomics study female ALD patients," female ALD patients over the age of 40 were stratified by disease severity into two groups, resulting in 26 patients in the "Mild" category and 28 patients in the "Severe" category.

For targeted LPC(26:0) analysis, an extra sample from a female with cerebral ALD was included, raising the total number of female samples in the study to 66. This inclusion was clarified in the Results section, titled "Plasma LPC(26:0) concentration determined by targeted assay correlates with disease severity in ALD patients," with the statement: "Additionally, we obtained and included plasma from a second female ALD patient with cerebral ALD," to address the discrepancy in the sample count. For targeted LPC(26:0) analysis on male ALD patients we were able to include an additional 20 patients raising the total number of male samples in the study to 112.

3.2

Were the samples were collected at the same facility and measured using the same equipment, or were they measured separately at three locations, and only the values were added together?

Response: Samples were measured at the Amsterdam UMC using the same equipment. We have updated the "Patient Selection and Clinical Assessment" section to include the following:

"All biochemical assays were performed at the Amsterdam UMC."

3.3

The categorization of patient groups into cases appears insufficient. In Fig. 2-4, cases are simply classified and analyzed based on the presence or absence of cerebral, adrenal, and spinal symptoms; however, since each patient may exhibit the presence or absence of each symptom, the categories could be divided into eight groups. Additionally, there might be a confounding effect among the symptoms of the cerebrum, adrenal glands, and spinal cord.

Alternatively, to analyze the effect of the cerebral symptoms on lipid molecular species, the following four categories could be defined: with both adrenal and spinal cord symptoms, with adrenal symptoms and without spinal symptoms, without adrenal symptoms with spinal cord symptoms and without both adrenal symptoms and spinal symptom. Within these four categories, the authors can compare the groups with and without cerebral symptoms. Identifying common changes in molecular species accompanying cerebral symptoms in all four categories would highlight it as a characteristic of cerebral symptoms. The same approach can be applied to adrenal symptoms and spinal symptoms.

Moreover, it is essential to clarify whether spinal symptoms are limited to patients without cerebral symptoms. It is generally known that spinal type AMN may transition to cerebral-type, and if a patient with cerebral-type consistently exhibits spinal symptoms, the patient classification could be differentiated based on the severity of central nervous system as cerebral symptoms type, spinal cord symptoms type only, or no symptoms type. These can be combined with or without adrenal symptoms, resulting in six groups.

Response: While the suggestion to stratify patients into smaller groups based on different combinations of symptoms may seem theoretically appealing, it poses practical challenges and risks losing meaningful effects within the data. Our rationale for the chosen stratification approach is based on knowledge of the natural history and the primary clinical syndromes observed: myelopathy, leukodystrophy, and adrenal insufficiency, which often coexist in the same patient. We have found an association between LPC(26:0) levels and these clinical syndromes. However, it's important to note that these associations are not independent. For example, elevated LPC(26:0) levels may indicate a higher risk of adrenal insufficiency, leukodystrophy, and spinal cord disease at the same time.

Furthermore, age has a significant impact on clinical outcomes, further complicating the stratification process. Dividing the cohort into numerous small groups may obscure rather than elucidate meaningful associations. Moreover, dividing patients into smaller groups reduces the sample size within each group, making it more difficult to identify significant differences between them. Our choice of grouping categories is intentional and is intended to capture the overall association between VLCFA-containing lipids and disease severity.

3.4

If the dataset does not include HCT except for specific cases, it should be explicitly mentioned.

Response: We agree with the reviewer and have updated the "Patient Selection and Clinical Assessment" section to include this information: "Unless specifically indicated, patients did not receive HCT."

3.5

The reason for creating the category for those over 55 is unclear. If age categorization is intended, it would be beneficial to subdivide the existing classification by age. Alternatively, analysis could be conducted exclusively for patients over 55 years of age in all categories. Even among those who have developed the disease, the values may be lower in people over 55 years of age. Naturally, controls should also be restricted to those over 55 years of age.

Response: Age stratification is based on knowledge of the natural history and is described in the patient stratification section:

“Patient stratification lipidomics study male ALD patients” In this study, our main objective was to find lipids that correlate with disease severity. To achieve this, we categorized male ALD patients based on the presence of cerebral ALD, adrenal insufficiency and severe spinal cord disease (EDSS >6) at assessment. As with all progressive disorders, a confounding factor is that individuals can develop new symptoms after the time of assessment. To mitigate this effect, we defined categories by using knowledge on natural history^{11,32,36}. For instance, if a male patient has normal adrenal function at an age of 55 years or later, it is likely to remain normal¹¹. Similarly, leukodystrophy only rarely occurs after the age of about 55 years. Severity of spinal cord disease is associated with age of onset, those with an EDSS score ≤6 in late adulthood are likely to remain ambulatory. Therefore, age (over 55 years) was used in addition to symptoms for final classification. Table 1 provides an overview of the different patient categories that were used in the lipidomics analysis.”

We also assessed LPC(26:0) concentration in 1090 healthy controls (both males and females) aged 0 to 78 years. Our findings show no age-related effects (Figure 6B).

3.6

The absence of a description, along with references, detailing the lipidomics methodology significantly undermines the credibility of the paper. The reliability of this research hinges on the precise separation and quantification of individual lipid molecular species, raising concerns about the adequacy of the current methodology. It is essential to describe the methods of extraction, separation including column and solvents, detection and identification method including ion mode and the mode of mass spectrometer. From the figure, it appears that a wide variety of lipid classes were targeted; however what ranges of lipid classes and fatty acid chain lengths and number of unsaturation bonds were targeted? What are the internal standards? If measurements were conducted separately in the three countries, the authors should describe each measurement method and elucidate the measures taken to ensure consistency and guarantee identical values from the same sample.

Response: All samples were measured at the Amsterdam UMC using the same equipment. We have updated the "Patient Selection and Clinical Assessment" section to include the following:

“All biochemical assays were performed at the Amsterdam UMC.”

We also have expanded the method section to include the sample preparation, analytical procedures and bioinformatics for the lipidomics analysis:

“Lipidomics analysis”

Lipidomics analysis was performed as previously described^{14,37}. In a 2 ml tube 20 µl of plasma was added and mixed with a mix of internal standards for different lipid classes, including 0.1 nmol of cardiolipin CL(14:0/14:0/14:0/14:0), 2.0 nmol of phosphatidylcholine PC(14:0/14:0), 0.1 nmol of phosphatidylglycerol PG(14:0/14:0), 5.0 nmol of phosphatidylserine PS(14:0/14:0), 0.5 nmol of phosphatidylethanolamine PE(14:0/14:0), 0.5 nmol of phosphatidic acid PA(14:0/14:0), 2.125 nmol of sphingomyelin SM(d18:1/12:0), 0.02 nmol of lysophosphatidylglycerol LPG(14:0), 0.1 nmol of lysophosphatidylethanolamine LPE(14:0), 0.5 nmol of lysophosphatidylcholine LPC(14:0), 0.1 nmol of lysophosphatidic acid LPA(14:0), 0.5 nmol of phosphatidylinositol PI(8:0/8:0), 0.5 nmol diglycerides DG(14:0/14:0), 0.5 nmol triglycerides TG(14:0/14:0/14:0), 2.5 nmol cholesterol ester D₇-CE(16:0), 0.125 nmol of sphingosine and ceramide mix (Avanti Polar Lipids) dissolved in 1:1 (v/v) methanol:chloroform. Next, 1.5 ml 1:1 (v/v) methanol:chloroform was added to each sample. The mixture was sonicated in a water bath (5 min) and centrifuged (4 °C, (16,000×g, 10 min). The supernatant was transferred to a 1.5 ml glass auto sampler vial and evaporated under a stream of nitrogen at 45°C. The dried lipids were reconstituted in 100 µl of 1:1 (v/v) chloroform:methanol. Chromatographic separation of lipids was done using a Thermo Fisher Scientific Ultimate 3000 binary UPLC using a normal phase and a reverse phase column in separate runs. Normal-phase separation was done using a Phenomenex® LUNA silica, 250×2 mm, 5 µm 100 Å column. Column temperature was held constant at 25°C. The composition of the mobile phase A consisted of 85:15 (v/v) methanol:water containing 0.0125% formic acid and 3.35 mmol/l ammonia and the composition of

mobile phase B consisted of 97:3 (v/v) chloroform:methanol containing 0.0125% formic acid. The LC gradient started at of 10% A for 0-1 min, 20% A at 4 min, 85% A at 12 min, 100% A at 12.1 min, 100% A for 12.1-14 min, 10% A at 14.1 min, 10% A for 14.1-15 min using a flow rate of 0.3 ml/min. Reversed-phase separation was done using a Waters HSS T3 column (150×2.1 mm, 1.8 μm particle size). The composition of the mobile phase A consisted of 4:6 (v/v) methanol:water and B 1:9 (v/v) methanol:isopropanol, both containing 0.1% formic acid and 10 mmol/l ammonia. The gradient started at 100% A going to 80% A at 1 min and 0% A at 16 min, 0% A for 16-20 min, 100% A at 20.1 min and 100% A for 20.1-21 min. The column temperature was held constant at 60°C and a flow rate of 0.4 ml/min was used. After LC separation, lipids were detected using a Q Exactive Plus Orbitrap mass spectrometer (Thermo Scientific) using negative and positive ionization. The spray voltage was 2,500 V and nitrogen was used as the nebulizing gas. A resolution of 280,000 was used in a mass range of m/z 150 to m/z 2,000.

Bioinformatics and statistics

The raw LC/MS data were converted to mzXML format using MSConvert⁴². Lipidomics data processing and analysis was done as described earlier³⁷. Briefly, lipidomics data were analyzed using an in-house developed lipidomics pipeline based on the R programming language (<http://www.r-project.org>) and MATLAB. Preprocessing was done using the R package XCMS with minor changes to some functions to better suit the Q Exactive™ data; notably, the definition of noise level in centWave was adjusted and the stepsize in fillPeaks⁴³. Lipid identification was based on a combination of accurate mass, (relative) retention times, analysis of samples with known metabolic defects, and the injection of relevant standards. Lipid classes were defined in our lipidomics pipeline in terms of their generic chemical formula, where R represents the radyl group. Upon import of the lipid database in the annotation pipeline, the generic chemical formula of each lipid class was expanded by replacing the R-element with a range of possible radyl group lengths and amount of unsaturations/double bonds. The resulting expanded list of chemical formulas was then used to calculate the neutral monoisotopic mass of each species. The monoisotopic mass was subsequently transformed to a set of m/z values for each adduct/charge combination with which that specific species could be reliably measured/annotated. The reported lipid abundances are semi-quantitative and calculated by dividing the response of the analyte (area of the peak) by that of the corresponding internal standard multiplied by the concentration of that internal standard (arbitrary unit, A.U)."

3.7

A comparison between the non-symptomatic group and the control is required.

Response: As requested by the reviewer, we added 3 volcano plots for the following comparisons to the Supplementary Material Figure 1:

- noCALD >55 versus control
- noAI >55 versus control
- mild spinal cord disease (males) versus control

These comparisons demonstrate that these patient groups have a characteristic ALD profile with elevated levels of VLCFA containing lipids (mostly LPC, PC and TG) compared to control.

3.8

The vertical axis of the trend line is unspecified. Please provide the calculation formula. Additionally, how would the trend line appear regarding the unsaturated bond number?

Response: In lipidr, chain length plots are created by plotting (log₂) fold changes and the total chain lengths of lipids within a specific lipid class in an xy plot and visualizing the trend by a regression line (LOESS curve). This information is now added to the Bioinformatics and statistics section:

"Lipid saturation and chain length plots were generated using the R package lipidr version 2.15.1. In lipidr, chain length plots are created by plotting a regression line (LOESS curve) of the (log₂) fold changes and the total chain lengths of lipids within a specific lipid class."

Concerning the potential trends regarding unsaturated bond number, we refer to the heatmap representations like Figure 2D. Plotting only the count of double bonds on a trendline would omit crucial information about carbon chain length (which is crucial information in ALD). In contrast, a heatmap, such as Figure 2D, illustrates trends in double bonds while also preserving information on the total carbon chain length.

3.9

In the analysis of each symptom, it is imperative to explicitly list the characteristic molecular species in the volcano plot as a table. Furthermore, by performing a volcano plot for each group and the control could reveal molecular species specific to each group.

Response: We have included a supplementary table with p values and fold changes for each comparison. We did not find additional meaningful insights by comparing each group to the control.

3.10

Please provide a specification of the characteristics of molecular species for female patients, both in comparison to male patients as a whole and those with spinal symptoms.

Response: Male and female patients were measured in different lipidomics runs. This means that we cannot directly compare lipidomics results between male and females. However, we have included a supplementary table with p values and fold changes for each comparison. Male and female have a similar lipidomic patterns where VLCFA are incorporated mostly in LPC, PC and TG.

3.11

It is essential to categorize patients undergoing HCT based on the degree of symptom improvement. Furthermore, variations in other molecular species obtained in above 9 should be described.

Response: HCT is performed in patients who are presymptomatic (only early stage leukodystrophy on brain MRI). After successful transplantation, there is stabilization of cerebral white matter lesions without overt symptoms.

3.12

How was it determined that 32:0-LPC is the 1-acyl form? On the other hand, what about 26:0-LPC?

Response: 1-acyl and 2-acyl forms of LPC are chromatographically separated on the reversed phase column we use in lipidomics. LPC(26:0) is presented in the 1-acyl form. We have updated the lipidomics section and figures from LPC(26:0) to 1-acyl LPC(26:0).

3.13

I would like to see measurements and considerations regarding the amount of molecules, not just increases and decreases. Is there a change in the composition, rather than a tiny amount of molecular fluctuation? Here, if you are only performing relative measurements instead of absolute quantification, you should confine the analysis to within-class analysis. The reason for this is that comparisons of signal intensities between different lipid classes are meaningless because their ionization efficiencies are completely different.

Response:

- We agree with the reviewer that comparisons of signal intensities between different lipid classes are not meaningful. This is why in we only compare levels of lipids from the same lipid class.
- Regarding the change in the total composition, it is important to realize that VLCFA containing lipids are relatively low abundant species compared to lipids with "normal" chain lengths. For example in plasma LPC(26:0) is in the 0.5-1.0 $\mu\text{mol/L}$ range while several "normal" length LPC's such as LPC(16:0) and LPC(18:0) are in the 10-50 $\mu\text{mol/L}$ range. Changes in VLCFA lipid levels would therefore have little effect on the total amount of LPC lipids.

- Changes in VLCFA levels between groups can span foldchanges of >2. We consider this not tiny fluctuations but significant in the context of ALD.
- We fully agree with the reviewer that absolute quantification has benefits compared to relative measurements. This is why we supported our lipidomics findings with the absolute quantification of LPC(26:0) using a validated diagnostic assay. Furthermore, we also performed absolute quantification of VLCFA using a targeted assay. As described in the manuscript, the results from both these assays align well with our lipidomics findings.

Reviewers' comments:

Reviewer #1 (Remarks to the Author):

The authors have addressed majority of my comments satisfactorily.

For comment 1.3 supporting publications include PMID: 14678759; 15353207; 18291369 etc should be referenced.

Reviewer #2 (Remarks to the Author):

The authors have satisfactorily addressed most of my concerns. Below are my minor comments regarding their responses:

2.7

Indeed, $\log -1$ and 1 translate to fold changes of 0.5 and 2 , respectively; however, these changes are generally not considered significant in the field of biomarker discovery, where $|\log| > 2-2.5$ is the usual threshold. For instance, the LPC C26:0 shows a 5-6 fold increase in ALD (male) patients compared to controls. It would be challenging to argue for the clinical significance of such fold changes.

The necessity for a stricter FDR correction arises from the data being nonrandom, rather than the opposite. The goal is not merely to identify any potential findings (thus minimizing type II errors) but to substantiate the compelling hypothesis presented here regarding the increase of VLCFA content with disease severity. Therefore, minimizing type I errors becomes crucial, when using lipidomic as supportive data.

The inherent challenges within lipidomics is well acknowledged, which has not yet established itself as a robust methodology. Nevertheless, I believe that the emphasis on the contributions of PC(48:4) and, in particular, LPC 32:0, should be reduced. They appear more as trends rather than correlations.

2.8 and 2.9

To refine my earlier question/comment: the average age of the nonCALD individuals in your cohort is 40. At this age, do you have an estimate of the percentage that may eventually progress to cALD? Presumed to be around 10%, given that this is merely 15 years below your selected cutoff age. Based on my small patient cohort and understanding of this disorder, transitioning to cALD after age 40 is uncommon, though I lack precise estimates (as noted in my 2.9 comment) and experts in the field may know better (e.g., the authors' group at Amsterdam UMC).

Therefore, theoretically, considering the possibility of some individuals transitioning, the 'actual' difference between CALD and nonCALD might be approximately 10% greater than what was observed, and not more than that. Does this percentage (whether 10 or a different value) align with the findings (nonALD and nonADL>55)?

Combined with point 2.7, these observations lead me to suggest that the strength of the findings could be moderated.

e.g., "The lipidome of the CALD group exhibited various significantly elevated lipids compared to the NoCALD group," - statistical significance.

"Notably, the fold increase in CALD compared to NoCALD became larger and more significant with increasing total acyl-chain length, as depicted in Figure 2C for LPC, PC, and TG," - indicates a trend.

2.13

While it is well recognized that females exhibit less pronounced VLCFA abnormalities, there is a distinction between demonstrating this in PCA analysis (figure 6A, lower panel) and asserting "PCA and PLS-DA indicate a distinct lipidome in female patients compared to controls (Figure 5A)." Labeling it as a 'distinct lipidome' may not be accurate, as it would be challenging to classify a sample based on this analysis alone.

Reviewer #3 (Remarks to the Author):

Although many of the noted issues have been improved, significant issues remain and cannot be determined as acceptable for publication as is.

Major Point

The description of possible confounding needs improvement. The symptoms observed in the cerebrum, adrenal gland, and spinal cord (Figs. 2-4, respectively) are more likely to be associated with confounding (factors).

From Table 1, the number of cases can be calculated as follows

CALD and AI; 19

CALD and non AI; 5

non CALD and AI; 31

non CALD and non AI; 37

>55

CALD and AI; 0

CALD and non AI; 2

non CALD and AI; 3

non CALD and non AI; 18

However, it is not specified how the number of spinal cord symptoms is included in the reported case counts. It is necessary that the above classification should be further divided into cases with and without spinal cord symptoms and the exact number of each should be clearly stated.

Minor Point

The author needs to add ">55" to the control in the legend of Supplemental Figure 1.

Re: Manuscript Number: COMMSMED-23-0845A

Lipidomic biomarkers in plasma correlate with disease severity in adrenoleukodystrophy.

We would like to thank the reviewers once again for their evaluation of the manuscript and the changes made based on their valuable comments. Below, we provide a point-by-point response and indicate whether changes have been made to sections of the manuscript.

First-round changes are highlighted in yellow and the second-round changes are highlighted in green.

Reviewers' comments:

Reviewer #1 (Remarks to the Author):

The authors have addressed majority of my comments satisfactorily. For comment 1.3 supporting publications include PMID: 14678759; 15353207; 18291369 etc should be referenced.

Response:

We thank the reviewer for indicating which publications should be added to the manuscript. As requested, we have added the reference PMID 14678759 (Correlation of very long chain fatty acid accumulation and inflammatory disease progression in childhood X-ALD) to the statement in the Introduction section: "Various studies show that excessive VLCFA in ALD are incorporated into complex lipids like phosphatidylcholines, cholesterol esters, and triglycerides²⁰⁻²³, likely driving neurodegeneration".

However, we did not include the references 15353207 and 18291369. This is because these two interesting studies focus on the effect of lovastatin treatment on an experimental autoimmune encephalomyelitis (EAE) rat model (15353207) and the effect of N-acetyl cysteine (NAC) on LPS-induced neuroinflammation.

Reviewer #2 (Remarks to the Author):

The authors have satisfactorily addressed most of my concerns. Below are my minor comments regarding their responses:

2.7

Indeed, log -1 and 1 translate to fold changes of 0.5 and 2, respectively; however, these changes are generally not considered significant in the field of biomarker discovery, where $|\log| > 2-2.5$ is the usual threshold. For instance, the LPC C26:0 shows a 5-6 fold increase in ALD (male) patients compared to controls. It would be challenging to argue for the clinical significance of such fold changes.

The necessity for a stricter FDR correction arises from the data being nonrandom, rather than the opposite. The goal is not merely to identify any potential findings (thus minimizing type II errors) but to substantiate the compelling hypothesis presented here regarding the increase of VLCFA content with disease severity. Therefore, minimizing type I errors becomes crucial, when using lipidomic as supportive data.

The inherent challenges within lipidomics is well acknowledged, which has not yet

established itself as a robust methodology. Nevertheless, I believe that the emphasis on the contributions of PC(48:4) and, in particular, LPC 32:0, should be reduced. They appear more as trends rather than correlations.

Response:

We thank the reviewer for the feedback on our revised manuscript. Our conclusions are not based on lipidomics data alone. In addition to our lipidomics results, we performed targeted analyses on LPC(26:0) and VLCFA, both of which corroborate our findings and significantly reduce the likelihood of type 1 errors. As requested by the reviewer, we have now included Bonferroni and Holm corrections of the p-values in Supplementary Materials 2 and 4. It is noteworthy that even after these stringent corrections, VLCFA lipids still exhibit significantly elevated levels in cases of severe disease.

Regarding fold change thresholds for biomarker discovery: in our study, we report 155 lipids with fold changes greater than 2 and 71 lipids with fold changes greater than 2.5. This includes only fold changes that are statistically significant and does not include the ALD vs. control comparisons.

As for the specific lipids discussed, PC(48:4) is used as an illustrative example in the figures and is not discussed in the text. LPC(32:0) is discussed twice in the text, as it turned out to be the most distinctive marker, showing the greatest difference. Below are the changes we made as requested by the reviewer:

Section: Cerebral ALD in males is associated with higher levels of VLCFA-containing lipids

Original: “Notably, the fold increase in CALD compared to NoCALD became larger and more significant with increasing total acyl-chain length, as depicted in Figure 2C for LPC, PC, and TG.”

Changed to: “Notably, the fold increase in CALD compared to NoCALD indicated a trend with increasing total acyl-chain length, as shown in Figure 2C for LPC, PC, and TG. 1-acyl LPC(32:0) showed the most significant statistical difference in CALD compared to NoCALD.”

Section: Discussion

Original: “Notably, 1-acyl LPC(32:0), a lipidomics marker, exhibited more significant fold changes between different patient groups compared to LPC(26:0). Interestingly, targeted LPC(26:0) analysis using MSMS as described by Hubbard et al. could be modified to include 1-acyl LPC(32:0)”

Changed to: “Notably, in our lipidomic dataset, 1-acyl LPC(32:0) showed a more pronounced statistical difference compared to LPC(26:0) in different patient groups. Further research is needed to evaluate LPC(32:0) as a potential biomarker for ALD. Interestingly, the targeted analysis of LPC(26:0) by MSMS as described by Hubbard et al. could be modified to include 1-acyl LPC(32:0).”

2.8 and 2.9

To refine my earlier question/comment: the average age of the nonCALD individuals in your cohort is 40. At this age, do you have an estimate of the percentage that may eventually progress to cALD? Presumed to be around 10%, given that this is merely 15 years below your selected cutoff age. Based on my small patient cohort and understanding of this disorder, transitioning to cALD after age 40 is uncommon, though I lack precise estimates (as noted in my 2.9 comment) and experts in the field may know better (e.g., the authors' group at Amsterdam UMC).

Therefore, theoretically, considering the possibility of some individuals transitioning, the 'actual' difference between CALD and nonCALD might be approximately 10% greater than what was observed, and not more than that. Does this percentage (whether 10 or a different value) align with the findings (nonALD and nonADL>55)?

Combined with point 2.7, these observations lead me to suggest that the strength of the findings could be moderated.

e.g., "The lipidome of the CALD group exhibited various significantly elevated lipids compared to the NoCALD group," - statistical significance.

"Notably, the fold increase in CALD compared to NoCALD became larger and more significant with increasing total acyl-chain length, as depicted in Figure 2C for LPC, PC, and TG," - indicates a trend.

Response:

We would like to emphasize that the percentage of patients transitioning to CALD does not necessarily correspond to an equivalent percentage change in the difference between the groups. The actual impact on group differences depends on the VLCFA lipid levels of the patients who are transitioning

Regarding the transition to cerebral ALD, it can be estimated that 25% of patients who are CALD-free at age 40 are likely to develop CALD by age 55 (based on Huffnagel et al, 2019). In our cohort, this translates to possibly 13 patients transitioning to CALD (although the exact conversion rate is still not well known and the subject of current studies). If we assume that these patients are likely those with relatively high levels of VLCFA lipids, the difference between noCALD and CALD observed will resemble those seen in the comparison between the noCALD >55 and CALD. Whether it is indeed predominantly those with relatively high VLCFA lipid levels who are transitioning to CALD is of high interest and the topic of our future studies. Also, the effect of possibly incorrect group assignment (i.e. developing CALD in the future in part of those patients now classified as noCALD) actually strengthens the findings, as the patients "dilute" the differences between the two groups.

As requested by the reviewer, we made the following textual changes:

Section: Cerebral ALD in males is associated with higher levels of VLCFA-containing lipids:

Original: “The lipidome of the CALD group exhibited various significantly elevated lipids compared to the NoCALD group (Figures 2A and 2B).”

Changed to: “The lipidome of the CALD group exhibited various elevated lipids compared to the NoCALD group (Figures 2A and 2B).”

Section: Cerebral ALD in males is associated with higher levels of VLCFA-containing lipids

Original: “Notably, the fold increase in CALD compared to NoCALD became larger and more significant with increasing total acyl-chain length, as depicted in Figure 2C for LPC, PC, and TG.”

Changed to: “Notably, the fold increase in CALD compared to NoCALD indicated a trend with increasing total acyl-chain length, as shown in Figure 2C for LPC, PC, and TG. 1-acyl LPC(32:0) showed the most significant statistical difference in CALD compared to NoCALD.”

2.13

While it is well recognized that females exhibit less pronounced VLCFA abnormalities, there is a distinction between demonstrating this in PCA analysis (figure 6A, lower panel) and asserting "PCA and PLS-DA indicate a distinct lipidome in female patients compared to controls (Figure 5A)." Labeling it as a 'distinct lipidome' may not be accurate, as it would be challenging to classify a sample based on this analysis alone.

Response:

As requested by the reviewer, we have updated the text in the section “Disease severity in female ALD patients is correlated with VLCFA-lipid levels”:

Original: “PCA and PLS-DA indicate a distinct lipidome in female patients compared to controls (Figure 5A).”

Changed to: “PCA and PLS-DA demonstrate that ALD affects the lipidome of female patients (Figure 5A).”

Reviewer #3 (Remarks to the Author):

Although many of the noted issues have been improved, significant issues remain and cannot be determined as acceptable for publication as is.

Major Point

The description of possible confounding needs improvement. The symptoms observed in the cerebrum, adrenal gland, and spinal cord (Figs. 2-4, respectively) are more likely to be associated with confounding (factors).

From Table 1, the number of cases can be calculated as follows

CALD and AI; 19

CALD and non AI; 5

non CALD and AI; 31

non CALD and non AI; 37

>55

CALD and AI; 0

CALD and non AI; 2

non CALD and AI; 3

non CALD and non AI; 18

However, it is not specified how the number of spinal cord symptoms is included in the reported case counts. It is necessary that the above classification should be further divided into cases with and without spinal cord symptoms and the exact number of each should be clearly stated.

Minor Point

The author needs to add ">55" to the control in the legend of Supplemental Figure 1.

Response:

We thank the reviewer for the feedback on our revised manuscript. Our hypothesis is that elevated VLCFA lipid levels are a major driver of disease severity, manifesting in cerebral, adrenal, and spinal symptoms. We propose that VLCFA dysregulation is the precursor to these diverse manifestations, rather than the consequence. This means that symptoms often occur simultaneously.

In addition, it is important to recognize that conditions such as cerebral ALD, adrenal insufficiency, and spinal cord disease are not static. Patients may develop one or more of these symptoms over time, adding complexity to subgroup analyses. This dynamic aspect influences the categorization of patients in longitudinal studies, as the disease state may evolve over time.

In response to the feedback, we have included Supplementary Figure 2 in the manuscript. This figure illustrates LPC(26:0) levels across patient groups, categorized by the combined presence or absence of cerebral ALD and adrenal insufficiency. Additionally, we added the following text:

Section: Plasma LPC(26:0) concentration determined by targeted assay correlates with disease severity in ALD patients

“In addition, supplementary Figure 2 illustrates the LPC(26:0) levels in patient groups categorized by the combined presence or absence of cerebral ALD and/or adrenal insufficiency. The highest LPC(26:0) levels were observed in patients with both cerebral ALD and adrenal insufficiency. Conversely, patients without one or both of these conditions had lower LPC(26:0) levels.”

Regarding further subdivision of based on the presence or absence of spinal cord symptoms and age, our current sample size limits our ability to effectively analyze such subdivided groups. Further subdivision of our cohort would result in groups far too small ($n < 5$ for many groups) to detect any statistical differences. As more patients are included in the future, these subgroup analyses will be interesting to do and could yield more insight.

Reviewers' comments:

Reviewer #2 (Remarks to the Author):

The authors addressed all my comments and I find the revised manuscript to be suitable for publication.

Reviewer #3 (Remarks to the Author):

Since a patient may have multiple symptoms, including cerebral, adrenal, and spinal cord, the data from this study is statistically very close to being confounded. I agree with the first response of the authors that if the groups were strictly divided according to the presence or absence of cerebral, adrenal, and spinal symptoms, the numbers would be small and the analysis would be meaningless. To make this situation clear, I pointed out in the previous review that the number of patients with spinal cord symptoms should be reported. However, this has not been improved in this revised version. Therefore, I cannot judge that the paper can be published as it is.

Major points

1) For male patients with spinal cord symptoms, the number of patients divided into subgroups based on cerebral and adrenal symptoms should be described. For patients with spinal cord symptoms used in Table 1 and Figure 4, the number of patients divided into the four subgroups CALD and AI, CALD and non AI, non CALD and AI, and non CALD and non AI should be described in the Methods or figure legends. These are the same subgroups as in Supplementary Figure 2.

2) The number of patients in the four subgroups in Supplementary Figure 2 does not match the number of patients calculated from Table 1. The inconsistency needs to be resolved.

CALD and AI; The number of patients calculated from Table 1 is 19, and the data in Supplementary Figure 2 is 19. This is consistent.

CALD and non AI; The number of patients calculated from Table 1 is 5, and the data in Supplementary Figure 2 is 6. This is inconsistent.

non CALD and AI; The number of patients calculated from Table 1 is 31, and the data in Supplementary Figure 2 is 34. This is inconsistent.

non CALD and non AI; The number of patients calculated from Table 1 is 37, and the data in Supplementary Figure 2 is 37. This is consistent.

Re: Manuscript Number: COMMSMED-23-0845B

Lipidomic biomarkers in plasma correlate with disease severity in adrenoleukodystrophy.

We would like to thank the reviewers once again for their evaluation of the manuscript and the changes made based on their valuable comments. Below, we provide a point-by-point response and indicate whether changes have been made to sections of the manuscript. Third-round changes are highlighted in blue.

Reviewers' comments:

Reviewer #2 (Remarks to the Author):

The authors addressed all my comments and I find the revised manuscript to be suitable for publication.

Reviewer #3 (Remarks to the Author):

Since a patient may have multiple symptoms, including cerebral, adrenal, and spinal cord, the data from this study is statistically very close to being confounded. I agree with the first response of the authors that if the groups were strictly divided according to the presence or absence of cerebral, adrenal, and spinal symptoms, the numbers would be small and the analysis would be meaningless. To make this situation clear, I pointed out in the previous review that the number of patients with spinal cord symptoms should be reported. However, this has not been improved in this revised version. Therefore, I cannot judge that the paper can be published as it is.

Major points

- 1) For male patients with spinal cord symptoms, the number of patients divided into subgroups based on cerebral and adrenal symptoms should be described. For patients with spinal cord symptoms used in Table 1 and Figure 4, the number of patients divided into the four subgroups CALD and AI, CALD and non-AI, non CALD and AI, and non CALD and non AI should be described in the Methods or figure legends. These are the same subgroups as in Supplementary Figure 2.
- 2) 2) The number of patients in the four subgroups in Supplementary Figure 2 does not match the number of patients calculated from Table 1. The inconsistency needs to be resolved.
CALD and AI; The number of patients calculated from Table 1 is 19, and the data in Supplementary Figure 2 is 19. This is consistent.
CALD and non AI; The number of patients calculated from Table 1 is 5, and the data in Supplementary Figure 2 is 6. This is inconsistent.
non CALD and AI; The number of patients calculated from Table 1 is 31, and the data in Supplementary Figure 2 is 34. This is inconsistent.
non CALD and non AI; The number of patients calculated from Table 1 is 37, and the data in Supplementary Figure 2 is 37. This is consistent.

Response:

We thank the reviewer for the thorough review of our work and have made a concerted effort to address these points. To address the reviewer's request, we have expanded Supplementary Figure 2 and added Supplementary Figure 2B. Supplementary Figure 2B divides male ALD patients into all possible combinations of spinal cord disease (mild or severe) with or without cerebral ALD, and with or without adrenal insufficiency. As noted by the reviewer, it is difficult to perform statistical analyses on these subdivisions due to the low number of subjects per group. However, this figure provides the insight requested by the reviewer into the distribution of CALD and AI in patients with mild and severe spinal cord disease. In addition, we have included the number of patients in each group for Supplementary Figure 2A and Supplementary Figure 2B.

We also added a reference to Supplementary Figure 2B in the text:

Section: Plasma LPC(26:0) concentration determined by targeted assay correlates with disease severity in ALD patients

"In addition, Supplementary Figure 2 illustrates LPC(26:0) levels in patient groups categorized by the presence or absence of cerebral ALD and/or adrenal insufficiency (Supplementary Figure 2A) and spinal cord severity (Supplementary Figure 2B). The highest LPC(26:0) levels were observed in patients with both cerebral ALD and adrenal insufficiency."

Table 1 and Figure 4 are based on lipidomics data. We would like to emphasize that Supplementary Figures 2A and 2B are based on our targeted LPC(26:0) MSMS assay. This targeted analysis is an established and validated diagnostic assay for peroxisomal disorders. Because of its established reliability, this assay is best suited for these types of sub-analyses where sample sizes are relatively small. Another advantage of this assay is its flexibility in terms of batch size. This allowed us to include more samples in the targeted LPC(26:0) analysis compared to the lipidomics analysis. We expanded the text accordingly:

Section: Plasma LPC(26:0) concentration determined by targeted assay correlates with disease severity in ALD patients:

"LPC(26:0) analysis as determined by UPLC-MS/MS plays a key role in ALD diagnosis, ALD newborn screening, and in the assessment of VLCFA accumulation in other peroxisomal disorders³⁸. Compared to lipidomics, this method provides more accurate concentration values in a sample, allowing for better longitudinal comparison. We expanded the cohort to include an additional 20 male and 1 female patients and performed targeted LPC(26:0) analysis on plasma samples from the 112 male and 66 female patients included in this study, as shown in Figure 6A and Supplementary Data 5."

For clarity, we have included an overview of the samples used for the targeted LPC(26:0) analysis in Supplementary Data 5, similar to the sample overview provided for the lipidomics dataset in Table 1. All data, including group information, used to generate Supplementary Figures 2A and 2B are provided in Supplementary Data 5. We encourage the reviewer to refer to Supplementary Data 5 to review all data and group information used.

REVIEWERS' COMMENTS:

Reviewer #3 (Remarks to the Author):

In this version, the classification of CALD and AI symptoms in patients with spinal cord symptoms has been described in Supplementary Figure 2B, making the situation of confounding factors clearer.

The authors also noted that the number of cases in Supplementary Figure 2A was greater than the lipidomics analysis in the results.

Based on the above, all necessary information has been described in this manuscript. With these improvements, this manuscript is considered to be acceptable.

However, there is a minor point that must be corrected.

Minor point

1) In the upper table of Supplementary Data 5, in the Cerebral ALD column, 71 in the NoCALD (all ages) row and 21 in the NoCALD >55 row should both be 0.

Re: Manuscript Number: COMMSMED-23-0845C

Lipidomic biomarkers in plasma correlate with disease severity in adrenoleukodystrophy.

We would like to thank the reviewers once again for their evaluation of the manuscript and the changes made based on their valuable comments. Below, we provide a point-by-point response and indicate whether changes have been made to sections of the manuscript. Third-round changes are highlighted in blue.

Reviewers' comments:

Reviewer #3 (Remarks to the Author):

In this version, the classification of CALD and AI symptoms in patients with spinal cord symptoms has been described in Supplementary Figure 2B, making the situation of confounding factors clearer.

The authors also noted that the number of cases in Supplementary Figure 2A was greater than the lipidomics analysis in the results.

Based on the above, all necessary information has been described in this manuscript. With these improvements, this manuscript is considered to be acceptable.

However, there is a minor point that must be corrected.

Minor point

1) In the upper table of Supplementary Data 5, in the Crebral ALD column, 71 in the NoCALD (all ages) row and 21 in the NoCALD >55 row should both be 0.

Answer: we thank the reviewer. We have corrected this error